# DADAO: Decoupled Accelerated Decentralized Asynchronous Optimization

## Abstract

DADAO is a novel decentralized asynchronous stochastic first order algorithm to minimize a sum of $L$-smooth and $\mu$-strongly convex functions distributed over a time-varying connectivity network of size $n$. We model the local gradient updates and gossip communication procedures with separate independent Poisson Point Processes, decoupling the computation and communication steps in addition to making the whole approach completely asynchronous. Our method employs primal gradients and does not use a multi-consensus inner loop nor other ad-hoc mechanisms such as Error Feedback, Gradient Tracking, or a Proximal operator. By relating the inverse of the smallest positive eigenvalue $\chi_1^*$ and the effective resistance $\chi_2^*$ of our graph to a necessary minimal communication rate between nodes of the network, we show that our algorithm requires $\mathcal{O}(n\sqrt{\frac{L}{\mu}}\log(\frac{1}{\epsilon}))$ local gradients and only $\mathcal{O}(n\sqrt{\chi_1^*\chi_2^*}\sqrt{\frac{L}{\mu}}\log(\frac{1}{\epsilon}))$ communications to reach a precision $\epsilon$. If SGD with uniform noise $\sigma^2$ is used, we reach a precision $\epsilon$ with same speed, up to a bias term in $\mathcal{O}(\frac{\sigma^2}{\sqrt{\mu L}})$. This improves upon the bounds obtained with current state-of-the-art approaches, our simulations validating the strength of our relatively unconstrained method.

## 1 Introduction

With the rise of highly-parallelizable and connected hardware, distributed optimization for machine learning is a topic of significant interest holding many promises. In a typical distributed training framework, the goal is to minimize a sum of functions $(f_i)_{i \leq n}$ split across $n$ nodes of a computer network. A corresponding optimization procedure involves alternating local computation and communication rounds between the nodes. Also, spreading the compute load is done to ideally obtain a *linear speedup* in the number of nodes. In the decentralized setting, there is no central machine aggregating the information sent by the workers: nodes are only allowed to communicate with their neighbors in the network. In this setup, optimal methods (Scaman et al., 2017; Kovalev et al., 2021a) have been derived for synchronous first-order algorithms, whose executions are blocked until a subset (or all) nodes have reached a predefined state: the instructions must be performed in a specific order (*e.g.*, all nodes must perform a local gradient step before the round of communication begins), which is one of the locks limiting their efficiency in practice.

This work attempts to simultaneously address multiple limitations of existing decentralized algorithms while guaranteeing fast convergence rates. To tackle the synchronous lock, we rely on the continuized framework (Even et al., 2021a), introduced initially to allow asynchrony in a fixed topology setting: iterates are labeled with a continuous-time index *(in opposition to a global iteration count)* and performed locally with no regards to a specific global ordering of events. This is more practical while being theoretically grounded and simplifying the analysis. However, in Even et al. (2021a), gradient and gossip operations are still coupled: each communication along an edge requires the computation of the gradients of the two functions locally stored on the corresponding nodes and vice-versa. As more communications steps than gradient computations are necessary to reach an $\epsilon$ precision, even in an optimal framework (Kovalev et al., 2021a; Scaman et al., 2017), the coupling directly implies an overload in terms of gradient steps. Another limitation is the restriction to a fixed topology: in a more practical setting, connections between nodes should be allowed to disappear or new ones to appear over time. The procedures of Kovalev et al. (2021c); Li & Lin

(2021) are the first to obtain an optimal complexity in gradient steps while being robust to topological change. Unfortunately, synchrony is mandatory in their frameworks as they either rely on the Error-Feedback mechanism (Stich & Karimireddy, 2020) or the Gradient Tracking one (Nedic et al., 2017). Moreover, they both use an inner loop to control the number of gradient steps at the cost of significantly increasing the number of activated communication edges. To our knowledge, this is the first work to tackle those locks simultaneously.

In this paper, we propose a novel algorithm (DADAO: Decoupled Accelerated Decentralized Asynchronous Optimization) based on a combination of similar formulations to Kovalev et al. (2021a); Even et al. (2021b); Hendrikx (2022) in the continuized framework of Even et al. (2021a). We study:

$$\inf_{x \in \mathbb{R}^d} \sum_{i=1}^{n} f_i(x), \tag{1}$$

where each $f_i : \mathbb{R}^d \to \mathbb{R}$ is a $\mu$-strongly convex and $L$-smooth function computed in one of the $n$ nodes of a network. We derive a first-order optimization algorithm that only uses primal gradients and relies on a time-varying Point-wise Poisson Processes (P.P.P.s (Last & Penrose, 2017)) modeling of the communication and gradient occurrences, leading to accelerated communication and computation rates. Our framework is based on a simple fixed-point iteration and kept minimal: it only involves primal computations with an additional momentum term and works in both the Gradient and Stochastic Gradient Descent (SGD) settings. Thus, we do not add other cumbersome designs such as the Error-Feedback or Forward-Backward used in Kovalev et al. (2021a), which are intrinsically synchronous. While we do not consider the delays bound to appear in practice (we assume instantaneous communications and gradient computations), we show that the ordering of the gradient and gossip steps can be variable, removing the coupling lock.

Our contributions are as follows: **(1)** first, we propose the first primal algorithm with provable guarantees in the context of asynchronous decentralized learning with time-varying connectivity. **(2)** This algorithm reaches accelerated rates of communication and computations while not requiring ad-hoc mechanisms obtained from an inner loop. **(3)** Our algorithm also leads to an accelerated rate with SGD with a minor modification. **(4)** We propose a simple theoretical framework compared to concurrent works, and **(5)** we demonstrate its optimality numerically.

The structure of our paper is as follows: in Sec. 3.1, we describe our work hypothesis and our model of a decentralized environment, while Sec. 3.2 describes our dynamic. Sec. 3.3 states our convergence guarantees whose proofs are fully detailed in the Appendix. Then, Sec. 4.1 compares our work with its competitors, Sec. 4.2 explains our implementation of this algorithm, and finally, Sec. 4.3 verifies our claims numerically. All our experiments are reproducible, using PyTorch (Paszke et al., 2019) and our code can be found in Appendix.

## 2 RELATED WORK

Tab. 1 compares our contribution with other references to highlight the benefits of our method.

Table 1: This table shows the strength of DADAO compared to concurrent works. $n$ is the number of node, $|\mathcal{E}|$ the number of edges, $\frac{1}{\chi_1}$ the smallest positive eigenvalue of a fixed stochastic Gossip matrix, $\rho$ the eigengap and $\chi_2 \leq \chi_1$ the effective resistance. Note that under reasonable assumptions $\sqrt{\chi_1 \chi_2} n = \mathcal{O}(|\mathcal{E}|\sqrt{\rho})$ (Sec. 4.1). Async., Comm., Grad., M.-C. and Prox. stand respectively for Asynchrony, Communication steps, Gradient steps., Multi-consensus and Proximal operator.

| Method | Async. | Varying Topology | Decoupled | No Inner Loop (M.-C. or Prox.) | Primal Oracle | Total # Comm. | Total # Grad. |
|---|---|---|---|---|---|---|---|
| MSDA (Scaman et al., 2017) | ✗ | ✗ | ✗ | ✗ | ✗ | $\sqrt{\rho}|\mathcal{E}|\sqrt{\frac{L}{\mu}}$ | $n\sqrt{\frac{L}{\mu}}$ |
| AGT (Li & Lin, 2021) | ✗ | ✓ | ✗ | ✗ | ✓ | $\chi_1|\mathcal{E}|\sqrt{\frac{L}{\mu}}$ | $n\sqrt{\frac{L}{\mu}}$ |
| ADOM+ (Kovalev et al., 2021a) | ✗ | ✓ | ✗ | ✗ | ✓ | $\chi_1|\mathcal{E}|\sqrt{\frac{L}{\mu}}$ | $n\sqrt{\frac{L}{\mu}}$ |
| Continuized (Even et al., 2021a) | ✓ | ✗ | ✗ | ✓ | ✗ | $\sqrt{\chi_1\chi_2}n\sqrt{\frac{L}{\mu}}$ | $\sqrt{\chi_1\chi_2}n\sqrt{\frac{L}{\mu}}$ |
| ADFS (Hendrikx et al., 2021) | ✗ | ✗ | ✓ | ✗ | ✗ | $\sqrt{\rho}|\mathcal{E}|\frac{L}{\mu}$ | $n\sqrt{\frac{L}{\mu}}$ |
| TVR (Hendrikx, 2022) | ✓ | ✗ | ✓ | ✓ | ✓ | $\rho|\mathcal{E}|\frac{L}{\mu}$ | $n\frac{L}{\mu}$ |
| DADAO (ours) | ✓ | ✓ | ✓ | ✓ | ✓ | $\sqrt{\chi_1\chi_2}n\sqrt{\frac{L}{\mu}}$ | $n\sqrt{\frac{L}{\mu}}$ |

**Continuized and asynchronous algorithms.** We highly rely on the elegant continuized framework (Even et al., 2021a), which allows obtaining simpler proofs and brings the flexibility of asynchronous algorithms. However, in our work, we significantly reduce the necessary amount of gradient steps compared to Even et al. (2021a) while maintaining the same amount of activated edges. Another type of asynchronous algorithm can also be found in Latz (2021), yet it fails to obtain Nesterov's accelerated rates for lack of momentum. We note that Leblond et al. (2018) studies the robustness to delays yet requires a shared memory and thus applies to a different context than decentralized optimization. Hendrikx (2022) is a promising approach for modeling random communication on graphs yet fails to obtain acceleration in a neat framework that does not use inner-loops.

**Decentralized algorithms with fixed topology.** Scaman et al. (2017) is the first work to derive an accelerated algorithm for decentralized optimization, and it links the convergence speed to the Laplacian eigengap. The corresponding algorithm uses a dual formulation and a Chebychev acceleration (synchronous and for a fixed topology). Yet, as stated in Tab. 2, it still requires many edges to be activated. Furthermore, under a relatively flexible condition on the intensity of our P.P.P.s, we show that our work improves over bounds that depend on the spectral gap. An emerging line of work following this formulation employs the continuized framework (Even et al., 2020; 2021a;b), but is unfortunately not amenable to incorporating a time-varying topology by essence, as they rely on a coordinate descent scheme in the dual (Nesterov & Stich, 2017). Finally, we note that Even et al. (2021b) incorporates delays in their model, using the same technique as our work, yet transferring this robustness to another setting remains unclear. Reducing the number of communication has been studied in Mishchenko et al. (2022), only in the context of constant topology and without obtaining accelerated rates. Hendrikx et al. (2021) allows for fast communication and gossip rates yet requires a proximal step and synchrony between nodes to apply a momentum variable.

**Decentralized algorithms with varying topology.** We highlight that Kovalev et al. (2021a); Li & Lin (2021); Koloskova et al. (2020) are some of the first works to propose a framework for decentralized learning in the context of varying topology. However, they rely on inner-loops propagating variables multiple times through a network, which imposes complete synchrony and communication overheads. In addition, as noted empirically in Lin et al. (2015), inner-loops lead to a plateau effect. Furthermore, we note that Kovalev et al. (2021b); Salim et al. (2021) employ a formulation derived from Salim et al. (2020); Condat et al. (2022), casting decentralized learning as a monotonous inclusion, obtaining a linear rate thanks to a preconditioning step of a Forward-Backward like algorithm. However, being sequential by nature, these types of algorithms are not amenable to a continuized framework.

**Error feedback/Gradient tracking.** A major lock for asynchrony is the use of Gradient Tracking (Koloskova et al., 2021; Nedic et al., 2017; Li & Lin, 2021) or Error Feedback (Stich & Karimireddy, 2020; Kovalev et al., 2021b). Indeed, gradient operations are locally tracked by a running-mean variable which is updated at each gradient update. This is incompatible with an asynchronous framework as it requires communication between nodes. Furthermore, a multi-consensus inner-loop seems mandatory to obtain accelerated rates, again not desirable.

**Decoupling procedures** Decoupling subsequent steps of an optimization procedures traditionally leads to speed-ups (Hendrikx et al., 2021; Hendrikx, 2022; Belilovsky et al., 2020; 2021). This contrasts with methods which couple gradient and gossip updates, such that they happen in a predefined order, i.e. simultaneously (Even et al., 2021a) or sequentially (Kovalev et al., 2021a; Koloskova et al., 2020). In decoupled optimization procedures, inner-loops are not desirable because they require an external procedure that can be potentially slow and need a block-barrier instruction during the algorithm's execution (e.g., Hendrikx et al. (2021; 2019)).

## 3 Fast Asynchronous Algorithm for Time-Varying Networks

### 3.1 Gossip Framework

We consider the problem defined by Eq. 1 in a distributed environment constituted by $n$ nodes whose dynamic is indexed by a continuous time index $t \in \mathbb{R}^+$. Each node has a local memory and can compute a local gradient $\nabla f_i$, as well as elementary operations, in an instantaneous manner. As

said above, having no delay is less realistic, yet adding them also leads to significantly more difficult proofs whose adaptation to our framework remains largely unclear. Next, we will assume that our computations and gossip result from independent in-homogeneous piecewise constant P.P.P. with no delay. For the sake of simplicity, we assume that all nodes can compute a gradient at the same rate:

**Assumption 3.1** (Homogeneous gradient computations). *The gradient computations are re-normalized to fire independently at a rate of 1 computation per time unit. For the $i$-th worker, we write $N_i(t)$ the corresponding P.P.P. of rate 1, as well as $\mathbf{N}(t) = (N_i(t))_{i \leq n}$.*

Next, we model the bandwidth of each machine. For an edge $(i, j) \in \mathcal{E}(t)$, we write $M_{ij}(t)$ the P.P.P. with rate $\lambda_{ij}(t) \geq 0$. When this P.P.P. fires, both nodes can potentially share their local memories. The rate $\lambda_{ij}(t)$ is adjustable locally by machine $i$, which can decide to speed or slow down its local communication. While $\lambda_{ij}(t)$ and $\lambda_{ji}(t)$ may refer to different quantities, we highlight that this communication process is symmetric, and we denote by $\bar{\mathcal{E}}(t)$ the corresponding undirected graph. Given our notations, we note that if $(i, j) \notin \mathcal{E}(t)$, then the connexion between $(i, j)$ can be thought as a P.P.P. with intensity 0. We now introduce the instantaneous expected gossip matrix of our graph:

$$\Lambda(t) \triangleq \sum_{(i,j) \in \mathcal{E}(t)} \lambda_{ij}(t)(e_i - e_j)(e_i - e_j)^\mathsf{T}.$$

We also write $\mathbf{\Lambda}(t) \triangleq \sum_{(i,j) \in \mathcal{E}(t)} \lambda_{ij}(t)(\mathbf{e}_i - \mathbf{e}_j)(\mathbf{e}_i - \mathbf{e}_j)^\mathsf{T}$ its tensorized counter-part that will be useful for our proofs and defining our Lyapunov potential, and $\Lambda^+(t)$ its pseudo inverse. Following Scaman et al. (2017), we will further compare this quantity to the centralized gossip matrix:

$$\pi \triangleq \mathbf{I} - \frac{1}{n}\mathbf{1}\mathbf{1}^\mathsf{T} = \frac{1}{2n}\sum_{i,j}(\mathbf{e}_i - \mathbf{e}_j)(\mathbf{e}_i - \mathbf{e}_j)^\mathsf{T}.$$

We introduce the instantaneous connectivity of $\Lambda(t)$, as in Kovalev et al. (2021a):

$$\frac{1}{\chi_1(t)} \triangleq \inf_{x \perp \mathbf{1}, \|x\|=1} x^\mathsf{T}\Lambda(t)x.$$

We might also write $\chi_1[\Lambda(t)]$ to avoid confusion, depending on the context. Next, we introduce the maximal effective resistance of the network, as in Even et al. (2021a); Ellens et al. (2011):

$$\chi_2(t) \triangleq \frac{1}{2}\sup_{(i,j) \in \mathcal{E}(t)}(e_i - e_j)^\mathsf{T}\Lambda^+(t)(e_i - e_j).$$

We remind the following Lemma (proof in Appendix D.2), which will be useful to control $\chi_1(t), \chi_2(t)$ and compare our bounds with the ones making use of the spectral gap of a graph:

**Lemma 3.1** (Bound on the connectivity constants). *The spectrum of $\Lambda(t)$ is non-negative. Furthermore, we have $\chi_1(t) = +\infty$ iff $\bar{\mathcal{E}}(t)$ is not a connected graph. Also, if the graph is connected, then $\frac{n-1}{\operatorname{Tr}\Lambda(t)} \leq \min(\chi_1(t), \chi_2(t))$. Furthermore, we also have $2\chi_2(t)\inf_{(i,j) \in \mathcal{E}(t)} \lambda_{ij}(t) \leq 1$.*

The last part of this Lemma allows to bound $\chi_2(t)$ when no degenerated behavior on the graph's connectivity happens: we assume the networks do not contain arbitrarily slow communication edges. The following assumption is necessary to avoid oscillatory effects due to the variations of $\Lambda(t)$:

**Assumption 3.2** (Slowly varying graphs). *Assume that $\Lambda(t)$ is piece-wise constant on time intervals.*

In particular, it implies that each $\lambda_{ij}(t)$ is piece-wise constant. Next, following Kovalev et al. (2021a), we bound uniformly the connectivity to avoid degenerated effects:

**Assumption 3.3** (Strongly connected topology). *Assume that there is $\chi_1^* > 0$ such that $\chi_1(t) \leq \chi_1^*$.*

We might write this quantity $\chi_1^*[\Lambda]$ to stress the dependency in $\Lambda(t)$. From supra, it's clear that $\chi_2(t) \leq \chi_1(t)$ so that under 3.3, $\chi_2(t)$ is upper bounded by $0 < \chi_2^* \leq \chi_1^*$.

### 3.2 DYNAMIC TO OPTIMUM

Next, we follow a standard approach (Kovalev et al., 2021c;a; Salim et al., 2022; Hendrikx, 2022) for solving Eq. 1 (see Appendix B for details), leading to, for $0 < \nu < \mu$:

$$\inf_{x \in \mathbb{R}^d} \sum_{i=1}^n f_i(x) = \inf_{x \in \mathbb{R}^{n \times d}} \sup_{y,z \in \mathbb{R}^{n \times d}} \sum_{i=1}^n f_i(x_i) - \frac{\nu}{2}\|x\|^2 - \langle x, y \rangle - \frac{1}{2\nu}\|\pi z + y\|^2 .$$

For $F(x) = \sum_{i=1}^n f_i(x_i) - \frac{\nu}{2}\|x\|^2$, the saddle points $(x^*, y^*, z^*)$ of this Lagrangian, are given by:

$$\begin{cases} \nabla F(x^*) - y^* & = 0 \\ \frac{y^* + \pi z^*}{\nu} + x^* & = 0 \\ \pi z^* + \pi y^* & = 0 . \end{cases} \tag{2}$$

Contrary to Kovalev et al. (2021a), we do not employ a Forward-Backward algorithm, which requires both an extra-inversion step and additional regularity on the considered proximal operator. Not only does this condition not hold in this particular case, but this is not desirable in a continuized framework where iterates are not ordered in a predefined sequence and require a local descent at each instant. Another major difference is that no Error-feedback is required by our approach, which allows to unlock asynchrony while simplifying the proofs and decreasing the required number of communications. Instead, we show it is enough to incorporate a standard fixed point algorithm, *without any specific preconditioning* (see Condat et al. (2019)). We consider the following dynamic:

$$\begin{cases} dx_t = \eta(\tilde{x}_t - x_t)dt - \gamma(\nabla F(x_t) - \tilde{y}_t)\,d\mathbf{N}(t) \\ d\tilde{x}_t = \tilde{\eta}(x_t - \tilde{x}_t)dt - \tilde{\gamma}(\nabla F(x_t) - \tilde{y}_t)\,d\mathbf{N}(t) \\ d\tilde{y}_t = -\theta(y_t + z_t + \nu\tilde{x}_t)dt + (\delta + \tilde{\delta})(\nabla F(x_t) - \tilde{y}_t)d\mathbf{N}(t) \\ dy_t = \alpha(\tilde{y}_t - y_t)dt \\ dz_t = \alpha(\tilde{z}_t - z_t)dt - \beta \sum_{(i,j)\in\mathcal{E}(t)}(\mathbf{e}_i - \mathbf{e}_j)(\mathbf{e}_i - \mathbf{e}_j)^{\mathsf{T}}(y_t + z_t)dM_{ij}(t) \\ d\tilde{z}_t = \tilde{\alpha}(z_t - \tilde{z}_t)dt - \tilde{\beta} \sum_{(i,j)\in\mathcal{E}(t)}(\mathbf{e}_i - \mathbf{e}_j)(\mathbf{e}_i - \mathbf{e}_j)^{\mathsf{T}}(y_t + z_t)dM_{ij}(t) , \end{cases} \tag{3}$$

where $\nu, \tilde{\eta}, \eta, \gamma, \alpha, \tilde{\alpha}, \theta, \delta, \tilde{\delta}, \beta, \tilde{\beta}$ are undetermined parameters yet. As in Nesterov (2003), variables are paired to obtain a Nesterov acceleration. The variables $(x, y)$ allow decoupling the gossip steps from the gradient steps using independent P.P.P.s. Furthermore, the Lebesgue integrable path of $\tilde{y}_t$ does not correspond to a standard momentum, as in a continuized framework (Even et al., 2021a); however, it turns out to be a crucial component of our method. Compared to Kovalev et al. (2021a), no extra multi-consensus step needs to be integrated. Our formulation of an asynchronous gossip step is similar to Even et al. (2021a) which introduces a stochastic variable on edges; however, contrary to this work, our gossip and gradient computations are decoupled. In fact, we can also consider SGD (Bottou, 2010), by replacing $\nabla F(x)$ by an estimate $\nabla F(x, \xi)$, for $\xi \in \Xi$, some measurable space. We will need the following assumption on the bias and variance of the gradient:

**Assumption 3.4** (Unbiased gradient with uniform additive noise)**.** *We assume that:*

$$\mathbb{E}_\xi \nabla F(x, \xi) = \nabla F(x) ,$$

*and that its quadratic error is uniformly bounded by $\sigma > 0$:*

$$\mathbb{E}_\xi \|\nabla F(x, \xi) - \nabla F(x)\|^2 \leq \sigma^2 .$$

Next, for SGD use, we modify the three first lines of Eq. 3, replacing the stochastic terms $(\nabla F(x_t) - \tilde{y}_t)\,d\mathbf{N}(t)$ by $\int_\Xi (\nabla F(x_t, \xi) - \tilde{y}_t)\,d\mathbf{N}(t, \xi)$, see Appendix C for the complete dynamic. Simulating those SDEs (Arnold, 1974) can be efficiently done, as explained in Sec. 4.2 and Appendix H.

### 3.3 THEORETICAL GUARANTEES

We follow the approach introduced in Even et al. (2021a) for studying the convergence of the dynamic 3. To do so, we introduce $X \triangleq (x, \tilde{x}, \tilde{y}), Y \triangleq (y, z, \tilde{z})$ and the following Lyapunov potential:

$$\Phi(t, X, Y) \triangleq A_t d_F(x, x^*) + \tilde{A}_t\|\tilde{x} - x^*\|^2 + B_t\|y - y^*\|^2 + \tilde{B}_t\|\tilde{y} - y^*\|^2$$
$$+ C_t\|z + y - z^* - y^*\|^2 + \tilde{C}_t\|\tilde{z} - z^*\|_{\mathbf{\Lambda}(t)^+}^2 ,$$

where $A_t, \tilde{A}_t, B_t, \tilde{B}_t, C_t, \tilde{C}_t, D_t$ are non-negative functions to be defined. We will use this potential to control the trajectories of $X_t \triangleq (x_t, \tilde{x}_t, \tilde{y}_t), Y_t \triangleq (y_t, z_t, \tilde{z}_t)$, leading to the equivalent dynamic:

$$\begin{cases} dX_t = a_1(X_t, Y_t)dt + b_1(X_t)d\mathbf{N}(t) \\ dY_t = a_2(X_t, Y_t)dt + \sum_{(i,j)\in\mathcal{E}(t)} b_2^{ij}(Y_t)dM_{ij}(t)\,, \end{cases}$$

where $a_1, a_2, b_1 = (b_1^i)_i, (b_2^{ij})_{ij}$ are smooth functions. We prove the following in Appendix D.3.

**Theorem 3.2** (Gradient Descent). *Assume each $f_i$ is $\mu$-strongly convex and $L$-smooth. For any $\Lambda(t)$, assume 3.1-3.3, and that $\chi_1^*[\Lambda]\chi_2^*[\Lambda] \le \frac{1}{2}$. Then there exists some parameters for the dynamic Eq. 3 (given in App. H.2), such that for any initialization $x_0 \in \mathbb{R}^d$, and $\tilde{x}_0 = x_0, y_0 = \tilde{y}_0 = \nabla f(x_0), z_0 = \tilde{z}_0 = -\pi\nabla f(x_0)$, we get for $t \in \mathbb{R}^+$ and $f(x) = \sum_{i=1}^n f_i(x_i)$:*

$$\mathbb{E}[f(x_t)] - f(x^*) \le \big(2(f(x_0) - f(x^*)) + \frac{\mu}{8}\|x_0 - x^*\|^2\big)e^{-\frac{t}{8\sqrt{2}}\sqrt{\frac{\mu}{L}}}\,.$$

*Also, the expected number of gradients is $nt$ and the expected number of edges activated is:*

$$\mathbb{E}\big[\int_0^t \sum_{(i,j)\in\mathcal{E}(u)} \lambda_{ij}(u)\, du\big] = \frac{1}{2}\int_0^t \mathbb{E}[\mathrm{Tr}\,\Lambda(u)]\, du\,. \tag{4}$$

We can obtain the following corollary with a minor modification of our current proof:

**Corollary 3.2.1** (SGD with additive noise). *Under the same assumption as Thm. 3.2 as well as 3.4, for the SGD-dynamic Eq. 8, the same parameters as Thm. 3.2 allows to obtain for $t \in \mathbb{R}^+$:*

$$\mathbb{E}[f(x_t)] - f(x^*) \le \big(2(f(x_0) - f(x^*)) + \frac{\mu}{8}\|x_0 - x^*\|^2\big)e^{-\frac{t}{8\sqrt{2}}\sqrt{\frac{\mu}{L}}} + \frac{5\sigma^2}{\sqrt{\mu L}}\,.$$

Following Even et al. (2021a), $L$ allows to adjust the trace-off bias-variance of our descent.

## 4 PRACTICAL IMPLEMENTATION

### 4.1 EXPECTED COMPUTATIONAL COMPLEXITY

For a given graph $\mathcal{E}(t)$, multiple choices of $\Lambda(t)$ are possible and would still lead to accelerated and linear rates as long as the condition $2\chi_1^*[\Lambda]\chi_2^*[\Lambda] \le 1$ is verified. Thus, we discuss how to choose our instantaneous expected gossip matrix to compare to concurrent work. To get a precision $\epsilon$, $T = \mathcal{O}(\sqrt{\frac{L}{\mu}}\log(\frac{1}{\epsilon}))$ local gradient computations is required per machine. More details can be found in Appendix F on our methodology for comparing with other methods, and particularly the way to recover the order of magnitudes we mention. In the following, each algorithm to which we compare ourselves is parameterized by a Laplacian matrix with various properties. Systematically, for an update frequency $f$ and a family of Laplacians $\{\mathcal{L}_q\}_q$ (which can be potentially reduced to a single element) given by concurrent work, we will set:

$$\Lambda(t) = \sqrt{2\chi_1^*[\mathcal{L}]\chi_2^*[\mathcal{L}]}\mathcal{L}_{\lfloor tf\rfloor} \triangleq \lambda^*\mathcal{L}_{\lfloor tf\rfloor}\,, \tag{5}$$

where $\lambda^*$ can be understood as a lower bound on the instantaneous expected communication rate. In this case, $\Lambda(t)$ satisfies the conditions of Theorem 3.2 or Corollary 3.2.1. From a physical point of view, it allows us to relate the spatial quantities of our graphs to a necessary minimal communication rate between nodes of the network; see Appendix E for a discussion on this topic.

**Comparison with ADOM+.** In ADOM+ (Kovalev et al., 2021a), one picks $\chi_1^*[\mathcal{L}] \ge 1$ and $f = \chi_1^*[\mathcal{L}]$. Then, the number of gossip steps of our algorithm is at most:

$$\sqrt{\chi_1^*[\mathcal{L}]\chi_2^*[\mathcal{L}]}\sup_q \mathrm{Tr}(\mathcal{L}_q)\sqrt{\frac{L}{\mu}}\log(\frac{1}{\epsilon}) = \mathcal{O}(\sqrt{\chi_1^*[\mathcal{L}]\chi_2^*[\mathcal{L}]}n\sqrt{\frac{L}{\mu}}\log(\frac{1}{\epsilon}))$$

Then, the expected communication of ADOM+ is potentially substantially higher than ours (see Appendix F.1):

$$\sum_{t=1}^T \chi_1^*[\mathcal{L}]|\mathcal{E}(t)| \ge \mathcal{O}(\sqrt{\chi_1^*[\mathcal{L}]\chi_2^*[\mathcal{L}]}n\sqrt{\frac{L}{\mu}}\log(\frac{1}{\epsilon}))\,.$$

**Comparison with standard Continuized.** If $\mathcal{L}$ is a Laplacian picked such that $\text{Tr}\,\mathcal{L} = 2$ (thus $f = 0$), as in Even et al. (2021a), then Even et al. (2021a) claims that at least $\mathcal{O}(\sqrt{\frac{L}{\mu}}\log(\frac{1}{\epsilon})\sqrt{\chi_1^*[\mathcal{L}]\chi_2^*[\mathcal{L}]})$ gradient and gossip iterations are needed. The number of gossip iterations is the same as ours, yet, thanks to Lemma 3.1, the number of gradient iterations can be substantially higher without any additional assumptions, as $n - 1 \leq 2\sqrt{\chi_1^*[\mathcal{L}]\chi_2^*[\mathcal{L}]}$ (see Appendix F.2). Furthermore, the computations of Even et al. (2021a) still use the dual gradients and are for a fixed topology.

**Comparison with methods that depend on the spectral gap.** For instance, MSDA relies on a Tchebychev acceleration of the number of gossip steps (possible because Scaman et al. (2017) uses a fixed gossip matrix) which allows getting the number of edges activated of about $\sqrt{\rho^*}|\mathcal{E}|\sqrt{\frac{L}{\mu}}\log(\frac{1}{\epsilon})$, where $\rho^*$ is the spectral gap. For our algorithm, the number of gossip writes (with $f = 0$):

$$\sqrt{\frac{L}{\mu}}\log(\frac{1}{\epsilon})\sqrt{\chi_1^*[\mathcal{L}]\chi_2^*[\mathcal{L}]}\text{Tr}\,\mathcal{L} \leq \mathcal{O}(\sqrt{\rho^*}|\mathcal{E}|\sqrt{\frac{L}{\mu}}\log(\frac{1}{\epsilon})),$$

where the details of this bound can be found in Appendix F.3 and relies solely on an assumption on the ratio between the maximal and minimal weights in the Laplacian. We highlight that Scaman et al. (2017); Kovalev et al. (2021a) claimed that their respective algorithms are optimal because they study the number of computations and synchronized gossips on a worst-case graph; our claim is, *by nature* different, as we are interested in the number of edges fired rather than the number of synchronized gossip rounds. Tab. 2 predicts the behavior of our algorithm for various classes of graphs encoded via the Laplacian of a stochastic matrix. It shows that systematically, our algorithm leads to the best speed[1]. We note that the graph class depicted in the Tab. 2 were used as worst-case examples of Scaman et al. (2017); Kovalev et al. (2021a). The next section implements and validates our ideas.

Table 2: Complexity for various graphs using a stochastic matrix. We have, respectively for a star / line or cyclic / complete graph and the $d$-dimensional grid: $\chi_1^* = \mathcal{O}(1)$, $\rho^* = \mathcal{O}(n)$ / $\chi_1^* = \mathcal{O}(n^2)$, $\rho^* = \mathcal{O}(n^2)$, $\chi_2^* = \mathcal{O}(1)$ / $\chi_1^* = \mathcal{O}(1)$, $\rho^* = \mathcal{O}(1)$ / $\chi_1^* = \mathcal{O}(n^{2/d})$, $\rho^* = \mathcal{O}(n^{2/d})$, $\chi_2^* = \mathcal{O}(1)$.

| Method | # edges activated | | | | # total gradient iterations | | | |
|---|---|---|---|---|---|---|---|---|
| Graph | Star | Line | Complete | $d$-grid | Star | Line | Complete | $d$-grid |
| (Kovalev et al., 2021a) ADOM+ | $n$ | $n^3$ | $n^2$ | $n^{1+2/d}$ | $n$ | $n$ | $n$ | $n$ |
| (Scaman et al., 2017) MSDA | $n^{3/2}$ | $n^2$ | $n^2$ | $n^{1+1/d}$ | $n$ | $n$ | $n$ | $n$ |
| (Even et al., 2021a) Continuized | $n$ | $n^2$ | $n$ | $n^{1+1/d}$ | $n$ | $n^2$ | $n$ | $n^{1+1/d}$ |
| Centralized | $n$ | - | - | - | $n$ | - | - | - |
| DADAO (ours) | $n$ | $n^2$ | $n$ | $n^{1+1/d}$ | $n$ | $n$ | $n$ | $n$ |

### 4.2 ALGORITHM

We now describe the algorithm used to implement the dynamics of Eq. 3 and, in particular, our simulator of P.P.P.. Let us write $T_1^{(i)} < T_2^{(i)} < ... < T_k^{(i)} < ...$ the time of the $k$-th event on the $i$-th node, which is either an edge activation, either a gradient update. We remind that the spiking times of a specific event correspond to random variables with independent exponential increments and can thus be generated at the beginning of our simulation. They can also be generated on the fly and locally to stress the locality and asynchronicity of our algorithm. Let's write $X_t = (X_t^{(i)})_i$ and $Y_t = (Y_t^{(i)})_i$, then on the $i$-th node and at the $k$-th iteration, we integrate the linear Ordinary Differential Equation (ODE) on $[T_k^{(i)}; T_{k+1}^{(i)}]$, given by $\begin{cases} dX_t = a_1(X_t, Y_t)dt \\ dY_t = a_2(X_t, Y_t)dt \end{cases}$, to define the values right before the spike, for $\mathcal{A}$ the corresponding constant matrix, we thus have:

$$\begin{pmatrix} X_{T_{k+1}^{(i)}-}^{(i)} \\ Y_{T_{k+1}^{(i)}-}^{(i)} \end{pmatrix} = \exp\left((T_{k+1}^{(i)} - T_k^{(i)})\mathcal{A}\right)\begin{pmatrix} X_{T_k^{(i)}}^{(i)} \\ Y_{T_k^{(i)}}^{(i)} \end{pmatrix}. \tag{6}$$

---

[1]For the case 2-grid, the logarithmic term should appear, yet we decided to neglect them.

Next, if one has a gradient update, then:

$$X^{(i)}_{T^{(i)}_{k+1}} = X^{(i)}_{T^{(i)-}_{k+1}} + b_1 \left( X^{(i)}_{T^{(i)-}_{k+1}} \right) .$$

Otherwise, if the edge $(i, j)$ or $(j, i)$ is activated, a communication bridge is created between both nodes $i$ and $j$. In this case, the local update on $i$ writes:

$$Y^{(i)}_{T^{(i)}_{k+1}} = Y^{(i)}_{T^{(i)-}_{k+1}} + b_2 \left( Y^{(i)}_{T^{(i)-}_{k+1}}, Y^{(j)}_{T^{(i)-}_{k+1}} \right) .$$

Note that, even if this event takes place along an edge $(i, j)$, we can write it separately for nodes $i$ and $j$ by making sure they both have the events $T^{(i)}_{k_i} = T^{(j)}_{k_j}$, for some $k_i, k_j \in \mathbb{N}$, corresponding to this communication. As advocated, all those operations are local, and we summarize in the Alg. 1 the algorithmic block which corresponds to our implementation. See Appendix H for more details.

---

**Algorithm 1:** This algorithm block describes our implementation on each local machine. The $ODE$ routine is described by Eq. 6 and Ping is an instantaneous routine.

**Input:** On each machine $i \in \{1, ..., n\}$, gradient oracle $\nabla f_i$, parameters $\mu, L, \chi_1^*, t_{\max}$.
1 **Initialize** on each machine $i \in \{1, ..., n\}$:
2     Set $X^{(i)}, Y^{(i)}, T^{(i)}$ to 0 and set $\mathcal{A}$ via Eq. 105;
3 **Synchronize** the clocks of all machines ;
4 **In parallel** *on workers* $i \in \{1, ..., n\}$, **while** $t < t_{\max}$, **continuously do:**
5     $t \leftarrow clock()$ ;
6     Ping surrounding machines and adjust $\lambda_{ij}(t)$ ;
7     **if** *there is an event at time* $t$ **then**
8        $(X^{(i)}, Y^{(i)}) \leftarrow ODE(\mathcal{A}, t - T^{(i)}, (X^{(i)}, Y^{(i)}))$ ;
9        **if** *the event is to take a gradient step* **then**
10           $X^{(i)} \leftarrow X^{(i)} + b_1(X^{(i)})$ ;
11        **else if** *the event is to communicate with* $j$ **then**
12           $Y^{(i)} \leftarrow Y^{(i)} + b_2(Y^{(i)}, Y^{(j)})$ ;          // Happens at $j$ simultaneously.
13        $T^{(i)} \leftarrow t$ ;
14 **return** $(x^{(i)}_{t_{\max}})_{1 \leq i \leq n}$, *the estimate of* $x^*$ *on each worker* $i$.

---

### 4.3 NUMERICAL RESULTS

In this section, we study the behavior of our method and compare it to several settings inspired by Kovalev et al. (2021a); Even et al. (2021a). In our experiments, we perform the empirical risk minimization for both the decentralized linear and logistic regression tasks given either by:

$$f_i(x) = \frac{1}{m} \sum_{j=1}^{m} \log(1 + \exp(-b_{ij} a_{ij}^\top x)) + \frac{\mu}{2} \|x\|^2 \quad \text{or} \quad f_i(x) = \frac{1}{m} \sum_{j=1}^{m} \|a_{ij}^\top x - c_{ij}\|^2, \quad (7)$$

where $a_{ij} \in \mathbb{R}^d$, $b_{ij} \in \{-1, 1\}$ and $c_{ij} \in \mathbb{R}$ correspond to $m$ local data points stored at node $i$. For both varying and fixed topology settings, we follow a protocol similar to Kovalev et al. (2021a): we generate $n$ independent synthetic datasets with the `make_classification` and `make_regression` functions of scikit-learn (Pedregosa et al., 2011), each worker storing $m = 100$ data points. We recall that the metrics of interest are the total number of local gradient steps and the total number of individual messages exchanged (i.e., *number of edges that fired*) to reach an $\epsilon$-precision. We systematically used the proposed hyper-parameters of each reference paper for our implementation without any specific fine-tuning.

**Comparison in the time-varying setting.** We compare our method to ADOM+ (Kovalev et al., 2021a) on a sequence of 50 random geometric graphs of size $n = 20$ in Fig. 1. To construct the graphs, we sample $n$ points uniformly in $[0, 1]^2 \subset \mathbb{R}^2$ and connect each of them to all at a distance less than some user-specified radius, which allows controlling the constant $\chi_1^*$ (we consider values in

$\{3, 33, 180, 233\}$). We ensure the connectedness of the graphs by adding a minimal number of edges, exactly as done in Kovalev et al. (2021b). We then use the instantaneous gossip matrix introduced in Eq. 5 with $f = \chi_1^*$. We compare ourselves to both versions of ADOM+: with and without the Multi-Consensus (M.-C.). Thanks to its M.-C. procedure, ADOM+ can significantly reduce the number of necessary gradient steps. However, consistently with our analysis in Sec. 4.1, our method is systematically better in all settings in terms of communications.

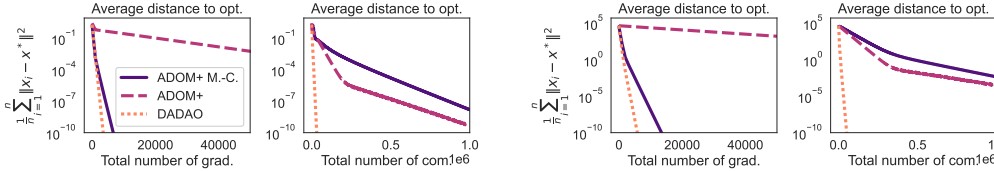

Figure 1: Comparison between ADOM+ (Kovalev et al., 2021a) and DADAO, using the same data *from left to right: binary classification, linear regression)* and the same sequence of random connected graphs with $\chi_1^* = 180$ linking $n = 20$ workers.

**Comparison with accelerated methods in the fixed topology setting.** Now, we fix the Laplacian matrix via Eq. 5 to compare simultaneously to the continuized framework (Even et al., 2021a) and MSDA (Scaman et al., 2017). We reports in Fig. 2 results corresponding to the complete graph with $n = 250$ nodes and the line graph of size $n = 150$. While sharing the same asymptotic rate, we note that the Continuized framework (Even et al., 2021a) and MSDA (Scaman et al., 2017) have better absolute constants than DADAO, giving them an advantage both in terms of the number of communication and gradient steps. However, in the continuized framework, the gradient and communication steps being coupled, the number of gradient computations can potentially be orders of magnitude worse than our algorithm, which is reflected by Fig. 2 for the line graph. As for MSDA and ADOM+, Tab. 2 showed they do not have the best communication rates on certain classes of graphs, as confirmed to the right in Fig. 2 for MSDA and the communication plots for ADOM+.

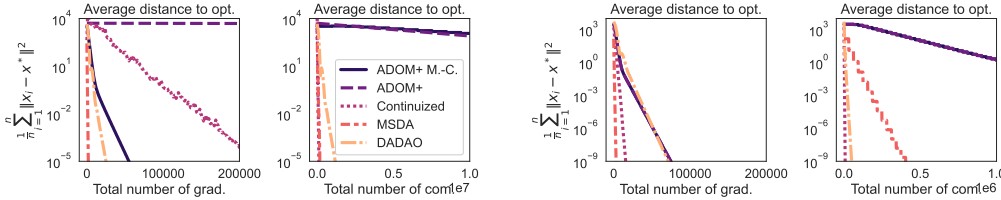

Figure 2: Comparison between ADOM+ (Kovalev et al., 2021a), the continuized framework (Even et al., 2021a), MSDA (Scaman et al., 2017) and DADAO, using the same data for the linear regression task, and the same graphs *(from left to right: line with $n = 150$, complete with $n = 250$).*

In conclusion, while several methods can share similar convergence rates, ours is the only one to perform at least as well as its competitors in every setting for different graph's topology and two distinct tasks, as predicted by Tab. 1.

## 5 CONCLUSION

In this work, we have proposed a novel stochastic algorithm for the decentralized optimization of a sum of smooth and strongly convex functions. We have demonstrated, theoretically and empirically, that this algorithm leads systematically to a substantial acceleration compared to state-of-the-art works. Furthermore, our algorithm is asynchronous, decoupled, primal, and does not rely on an extra inner-loop while being amenable to varying topology settings: each of these properties makes it suitable for real applications. In future work, we would like to explore the robustness of such algorithms to more challenging variabilities occurring in real-life applications.

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

## A  NOTATIONS

For a positive semi-definite matrix $A$, $\|x\|_A \triangleq x^\mathsf{T} A x$, $f = \mathcal{O}(g)$ means there is a constant $C > 0$ such that $|f| \leq C|g|$, $\{e_i\}_{i \leq d}$ is the canonical basis of $\mathbb{R}^d, d \in \mathbb{N}$, $\mathbf{1}$ is the vector of 1, $\mathbf{I}$ the identity, $A^+$ is the pseudo-inverse of $A$ and for a smooth convex function $F$,

$$d_F(x, y) \triangleq F(x) - F(y) - \langle \nabla F(y), x - y \rangle$$

is its Bregman divergence. We further write $\mathbf{e}_i \triangleq e_i \otimes \mathbf{I}$.

## B  SADDLE POINT REFORMULATION

With $0 < \nu < \mu$ and introducing an extra dual variable $\hat{x}$, we get:

$$
\begin{aligned}
\inf_{x \in \mathbb{R}^d} \sum_{i=1}^n f_i(x) &= \inf_{\substack{x,\hat{x} \in \mathbb{R}^{n \times d} \\ x=\hat{x}, \pi\hat{x}=0}} \sum_{i=1}^n f_i(x_i) - \frac{\nu}{2}\|x\|^2 + \frac{\nu}{2}\|\hat{x}\|^2 \\
&= \inf_{x,\hat{x} \in \mathbb{R}^{n \times d}} \sup_{y,z \in \mathbb{R}^{n \times d}} \sum_{i=1}^n f_i(x_i) - \frac{\nu}{2}\|x\|^2 + \frac{\nu}{2}\|\hat{x}\|^2 + \langle y, \hat{x} - x \rangle + \langle z, \pi\hat{x} \rangle \\
&= \inf_{x \in \mathbb{R}^{n \times d}} \sup_{y,z \in \mathbb{R}^{n \times d}} \inf_{\hat{x} \in \mathbb{R}^{n \times d}} \sum_{i=1}^n f_i(x_i) - \frac{\nu}{2}\|x\|^2 + \frac{\nu}{2}\|\hat{x}\|^2 + \langle y, \hat{x} - x \rangle + \langle z, \pi\hat{x} \rangle \\
&= \inf_{x \in \mathbb{R}^{n \times d}} \sup_{y,z \in \mathbb{R}^{n \times d}} \sum_{i=1}^n f_i(x_i) - \frac{\nu}{2}\|x\|^2 - \langle x, y \rangle - \frac{1}{2\nu}\|\pi z + y\|^2 .
\end{aligned}
$$

## C  SGD DYNAMIC

The dynamic considered when using stochastic gradients is given by:

$$
\begin{cases}
dx_t = \eta(\tilde{x}_t - x_t)dt - \gamma \int_\Xi (\nabla F(x_t, \xi) - \tilde{y}_t)\, d\mathbf{N}(t, \xi) \\
d\tilde{x}_t = \tilde{\eta}(x_t - \tilde{x}_t)dt - \tilde{\gamma} \int_\Xi (\nabla F(x_t, \xi) - \tilde{y}_t)\, d\mathbf{N}(t, \xi) \\
d\tilde{y}_t = -\theta(y_t + z_t + \nu\tilde{x}_t)dt + (\delta + \tilde{\delta}) \int_\Xi (\nabla F(x_t, \xi) - \tilde{y}_t) d\mathbf{N}(t, \xi) \\
dy_t = \alpha(\tilde{y}_t - y_t)dt \\
dz_t = \alpha(\tilde{z}_t - z_t)dt - \beta \sum_{(i,j) \in \mathcal{E}(t)} (\mathbf{e}_i - \mathbf{e}_j)(\mathbf{e}_i - \mathbf{e}_j)^\mathsf{T}(y_t + z_t)dM_{ij}(t) \\
d\tilde{z}_t = \tilde{\alpha}(z_t - \tilde{z}_t)dt - \tilde{\beta} \sum_{(i,j) \in \mathcal{E}(t)} (\mathbf{e}_i - \mathbf{e}_j)(\mathbf{e}_i - \mathbf{e}_j)^\mathsf{T}(y_t + z_t)dM_{ij}(t)
\end{cases}
\tag{8}
$$

## D  PROOF OF THE THEOREM

### D.1  PROPERTIES AND ASSUMPTIONS

The following properties will be used all along the proofs of the Lemmas and Theorems and are related to the communication of our nodes.

**Lemma D.1.** *Under the assumptions of Theorem 3.2, if $z_0, \tilde{z}_0 \in span(\pi)$, then $z_t, \tilde{z}_t \in span(\pi)$ almost surely.*

*Proof.* It's clear that for any $i, j$, we get:

$$\pi(\mathbf{e}_i - \mathbf{e}_j)(\mathbf{e}_i - \mathbf{e}_j)^\mathsf{T} = (\mathbf{e}_i - \mathbf{e}_j)(\mathbf{e}_i - \mathbf{e}_j)^\mathsf{T} .$$

Thus, the variations of $(z_t, \tilde{z}_t)$ belong to span$(\pi)$, and so is the trajectory. $\qquad \square$

We derive the following Lemma, similar to a result from Boyd et al. (2006):

**Lemma D.2** (Spiking contraction). *Under the assumptions of Theorem 3.2, we have:*

$$\sum_{(i,j)\in\mathcal{E}(t)} \lambda_{ij}(t)\big[\|(\mathbf{e}_i - \mathbf{e}_j)(\mathbf{e}_i - \mathbf{e}_j)^\mathsf{T}x - \pi x\|^2 - \|\pi x\|^2\big] = -x^\mathsf{T}\mathbf{\Lambda}(t)x \leq -\frac{1}{\chi_1^*}\|\pi x\|^2\,.$$

*Proof.* If $i = j$, then $\lambda_{ii} = 0$. For a given $(i,j)$, we get if $i \neq j$:

$$\|(\mathbf{e}_i - \mathbf{e}_j)(\mathbf{e}_i - \mathbf{e}_j)^\mathsf{T}x - \pi x\|^2 = \|\pi x\|^2 + \|x_i - x_j\|^2 \tag{9}$$
$$- 2\langle\pi(x), (\mathbf{e}_i - \mathbf{e}_j)(\mathbf{e}_i - \mathbf{e}_j)^\mathsf{T}x\rangle$$
$$= \|\pi(x)\|^2 - \langle x, (\mathbf{e}_i - \mathbf{e}_j)(\mathbf{e}_i - \mathbf{e}_j)^\mathsf{T}x\rangle\,. \tag{10}$$

And this allows conclusion by sum. $\qquad\square$

**Lemma D.3** (Effective resistance contraction). *For $i, j$ and any $x \in \mathbb{R}^d$, we have:*

$$\|(\mathbf{e}_i - \mathbf{e}_j)(\mathbf{e}_i - \mathbf{e}_j)^\mathsf{T}x\|_{\mathbf{\Lambda}(t)^+}^2 \leq \chi_2^*\|(\mathbf{e}_i - \mathbf{e}_j)(\mathbf{e}_i - \mathbf{e}_j)^\mathsf{T}x\|\,.$$

*Proof.* Indeed, we note that:

$$\|(\mathbf{e}_i - \mathbf{e}_j)(\mathbf{e}_i - \mathbf{e}_j)^\mathsf{T}x\|_{\mathbf{\Lambda}(t)^+}^2 = x^\mathsf{T}(\mathbf{e}_i - \mathbf{e}_j)(\mathbf{e}_i - \mathbf{e}_j)^\mathsf{T}\mathbf{\Lambda}(t)^+(\mathbf{e}_i - \mathbf{e}_j)(\mathbf{e}_i - \mathbf{e}_j)^\mathsf{T}x \tag{11}$$
$$\leq 2\chi_2^* x^\mathsf{T}(\mathbf{e}_i - \mathbf{e}_j)(\mathbf{e}_i - \mathbf{e}_j)^\mathsf{T}x \tag{12}$$
$$= \chi_2^*\|(\mathbf{e}_i - \mathbf{e}_j)(\mathbf{e}_i - \mathbf{e}_j)^\mathsf{T}x\|^2 \tag{13}$$

$$\square$$

**Lemma D.4** (Min and max voltage values, see, *e.g.*, Klein & Randic (1993).). *For any $i, j, k$, $e_i^\mathsf{T}\Lambda(t)^+(e_i - e_j) \geq e_k^\mathsf{T}\Lambda(t)^+(e_i - e_j)$.*

*Proof.* Let us call $v \triangleq \Lambda(t)^+(e_i - e_j)$, and for a vertex $k$, $N(k)$ the set its neighbors in $\mathcal{E}(t)$. Note that $k \notin N(k)$. We want to prove that $v_i \triangleq e_i^\mathsf{T}v$ is greater than $v_k$. In fact, we will prove that $\forall k, v_i \geq v_k \geq v_j$. Recall that $\Lambda(t)\Lambda(t)^+ = \mathbf{I} - \pi$, meaning that $\Lambda(t)v = e_i - e_j$. Thus, $\forall k \notin \{i, j\}$, we have $(\Lambda(t)v)_k = 0$, leading to:

$$0 = v_k \sum_{k'\in N(k)}(\lambda_{kk'}(t) + \lambda_{k'k}(t)) - \sum_{k'\in N(k)}(\lambda_{kk'}(t) + \lambda_{k'k}(t))v_{k'}\,.$$

This allows us to write:

$$v_k = \frac{\sum_{k'\in N(k)}(\lambda_{kk'}(t) + \lambda_{k'k}(t))v_{k'}}{\sum_{k'\in N(k)}\lambda_{kk'}(t) + \lambda_{k'k}(t)}\,,$$

meaning that $v_k$ is a convex combination of the values of $v$ of its neighbors. As such, for any $k \notin \{i, j\}$, the value $v_k$ is inside the convex hull of the $\{v_{k'}, k' \in N(k)\}$ and cannot be strictly superior to the maximal nor strictly inferior to the minimal value of $v$ among its neighbors. The only case where $v_k$ is maximal or minimal is when the $\{v_{k'}, k' \in N(k)\}$ are all equal. This means that the only two coordinates of $v$ that are allowed to be strictly maximal or minimal among their neighbors are $i$ and $j$. Thus, the graph being connected, there is a path beginning at any $k$ and leading to $i$ or $j$ made by iterating $\arg\max_{k'\in N(k)} v_{k'}$ or $\arg\min_{k'\in N(k)} v_{k'}$ steps, hence $\max_k v_k$ and $\min_k v_k$ are in $\{v_i, v_j\}$. But, we have $v_i - v_j = (e_i - e_j)^\mathsf{T}\Lambda(t)^+(e_i - e_j) \geq 0$ and $v_i \geq v_j$. Thus, $\max_k v_k = v_i$ and $\min_k v_k = v_j$. $\qquad\square$

**Lemma D.5** (Bound effective resistance). *For any $i, j$, we have $\lambda_{ij}(t)(e_i - e_j)^\mathsf{T}\Lambda(t)^+(e_i - e_j) \leq 1$.*

*Proof.* As $\Lambda(t)\Lambda(t)^+ = \mathbf{I} - \pi$, for any $i, j$, we write:

$$2 = (e_i - e_j)^\mathsf{T}\Lambda(t)\Lambda(t)^+(e_i - e_j) \tag{14}$$
$$= \sum_{(k,l)\in\mathcal{E}(t)}\lambda_{kl}(t)(e_i - e_j)^\mathsf{T}(e_k - e_l)(e_k - e_l)^\mathsf{T}\Lambda(t)^+(e_i - e_j)\,. \tag{15}$$

As we have, for any $i, j, k, l$:

$$(e_i - e_j)^\mathsf{T}(e_k - e_l) = \begin{cases} 1 & \text{if } k = i \text{ and } l \neq j \\ 1 & \text{if } k \neq i \text{ and } l = j \\ -1 & \text{if } k = j \text{ and } l \neq i \\ -1 & \text{if } k \neq j \text{ and } l = i \\ 2 & \text{if } k = i \text{ and } l = j \\ -2 & \text{if } k = j \text{ and } l = i \\ 0 & \text{otherwise} \end{cases}, \tag{16}$$

and applying the fact that $\lambda_{kl}(t) = 0$ if $(k,l) \notin \mathcal{E}(t)$, we can expand Eq. 15 as:

$$\begin{aligned} 2 = &\sum_{l \neq j} \lambda_{il}(t) \left( e_i^\mathsf{T} \Lambda(t)^+ (e_i - e_j) - e_l^\mathsf{T} \Lambda(t)^+ (e_i - e_j) \right) \\ &+ \sum_{k \neq i} \lambda_{kj}(t) \left( e_k^\mathsf{T} \Lambda(t)^+ (e_i - e_j) - e_j^\mathsf{T} \Lambda(t)^+ (e_i - e_j) \right) \\ &- \sum_{l \neq i} \lambda_{jl}(t) \left( e_j^\mathsf{T} \Lambda(t)^+ (e_i - e_j) - e_l^\mathsf{T} \Lambda(t)^+ (e_i - e_j) \right) \\ &- \sum_{k \neq j} \lambda_{ki}(t) \left( e_k^\mathsf{T} \Lambda(t)^+ (e_i - e_j) - e_i^\mathsf{T} \Lambda(t)^+ (e_i - e_j) \right) \\ &+ 2\lambda_{ij}(t)(e_i - e_j)^\mathsf{T} \Lambda(t)^+ (e_i - e_j) \\ &- 2\lambda_{ji}(t)(e_j - e_i)^\mathsf{T} \Lambda(t)^+ (e_i - e_j). \end{aligned} \tag{17}$$

Using Lemma D.4, the facts that $\lambda_{ji}(t) \geq 0$ and $\Lambda(t)^+$ is positive semi-definite, all the terms in Eq. 17 are positive, giving:

$$2\lambda_{ij}(t)(e_i - e_j)^\mathsf{T} \Lambda(t)^+ (e_i - e_j) \leq 2 \tag{18}$$

$\square$

Next, we set $\nu = \frac{\mu}{2}$ such that:

$$\frac{1}{2L} \|\nabla F(x) - \nabla F(y)\|^2 \leq d_F(x,y) \leq \frac{L}{2} \|x - y\|^2,$$

and

$$\frac{\nu}{2} \|x - y\|^2 \leq d_F(x,y) \leq \frac{1}{2\nu} \|\nabla F(x) - \nabla F(y)\|^2,$$

and we remind that:

$$\mathbb{E}_\xi d_{F(.,\xi)}(x,y) = d_{\mathbb{E}_\xi F(.,\xi)}(x,y). \tag{19}$$

### D.2 PROOF OF THE LEMMA 3.1

*Proof of Lemma 3.1.* First, we note that $\Lambda(t)$ is symmetric and has a non-negative spectrum, as:

$$x^\mathsf{T} \Lambda(t) x = \sum_{(ij) \in \mathcal{E}(t)} \lambda_{ij}(t) \|x_i - x_j\|^2.$$

From this, we also clearly see that $\chi_1(t) = +\infty$ iff the graph is disconnected. Next, assuming that the graph is connected, 0 is an eigenvalue of $\Lambda(t)$ with multiplicity 1 and by definition of $\chi_1(t)$, we have $\text{Tr}\Lambda(t) \geq \frac{n-1}{\chi_1(t)}$. As we also have:

$$\sum_{(i,j) \in \mathcal{E}(t)} \lambda_{ij}(t)(e_i - e_j)^\mathsf{T} \Lambda^+(t)(e_i - e_j) = \text{Tr}(\Lambda^+(t)\Lambda(t)) = n - 1,$$

we can write:

$$n - 1 \leq 2\chi_2(t) \sum_{(i,j) \in \mathcal{E}(t)} \lambda_{ij}(t) = \chi_2(t)\text{Tr}\Lambda(t)$$

and get $\frac{n-1}{\mathrm{Tr}\Lambda(t)} \leq \min(\chi_1(t), \chi_2(t))$. Finally, for any $(i,j) \in \mathcal{E}(t)$, using Lemma D.5, we get that:

$$\lambda_{ij}(t)(e_i - e_j)^\mathsf{T}\Lambda^+(t)(e_i - e_j) \leq 1\,.$$

Thus,

$$\left(\inf_{(i,j)\in\mathcal{E}(t)} \lambda_{ij}(t)\right)(e_i - e_j)^\mathsf{T}\Lambda^+(t)(e_i - e_j) \leq 1\,,$$

leading to:

$$2\chi_2(t) \leq \frac{1}{\inf_{(i,j)\in\mathcal{E}(t)} \lambda_{ij}(t)}$$

. $\qquad\qquad\square$

### D.3 Proof of Theorem 3.2 and Corollary 3.2.1

*Proof of Theorem 3.2.* Because $\Phi$ is smooth and $\mathcal{E}(t)$ is constant on intervals, we get via Ito's formula (Last & Penrose, 2017) applied to the semi-martingale $(X_t, Y_t)$, gluing intervals where $\mathcal{E}(t)$ is constant (as well as the weights $\lambda_{ij}(t)$), that:

$$\Phi(t, X_t, Y_t) = \Phi(0, X_0, Y_0) + \int_0^T \langle \nabla\Phi(t, X_t, Y_t), \begin{pmatrix} 1 \\ a_1(X_t, Y_t) \\ a_2(X_t, Y_t) \end{pmatrix} \rangle dt$$

$$+ \sum_{i=1}^n \int_0^T \left(\Phi(t, X_t + b_1^i(X_t), Y_t) - \Phi(t, X_t, Y_t)\right)dt$$

$$+ \sum_{(i,j)\in\mathcal{E}(t)} \int_0^T \left(\Phi(t, X_t, Y_t + b_2^{ij}(Y_t)) - \Phi(t, X_t, Y_t)\right)\lambda_{ij}(t)dt + \Theta_T\,,$$

where:

$$\Theta_T \triangleq \sum_{i=1}^n \int_0^T \left(\Phi(t, X_{t^-}, Y_{t^-} + b_1^i(X_{t^-})) - \Phi(u, X_{t^-}, Y_{t^-})\right)(dN_i(t) - dt)$$

$$+ \sum_{(i,j)\in\mathcal{E}(t)} \int_0^T \left(\Phi(t, X_{t^-} + b_2^{ij}(X_{t^-}), Y_{t^-}) - \Phi(t, X_{t^-}, Y_{t^-})\right)(dM_{ij}(t) - \lambda_{ij}(t)dt)\,.$$

We will use the following technical Lemma, which is also difficult to prove and whose proof is deferred to Appendix D.4:

**Lemma D.6.** *There exists some parameters $\nu, \tilde{\eta}, \eta, \gamma, \tilde{\gamma}, \alpha, \tilde{\alpha}, \theta, \delta, \tilde{\delta}, \beta, \tilde{\beta}$ and $c > 0$ such that:*

$$\langle \nabla\Phi(t, X_t, Y_t), \begin{pmatrix} 1 \\ a_1(X_t, Y_t) \\ a_2(X_t, Y_t) \end{pmatrix} \rangle + \left(\Phi(t, X_t + b_1(X_t), Y_t) - \Phi(t, X_t, Y_t)\right)$$

$$+ \sum_{(i,j)\in\mathcal{E}(t)} \lambda_{ij}(t)\left(\Phi(t, X_t, Y_t + b_2^{ij}(Y_t)) - \Phi(t, X_t, Y_t)\right) \leq 0 \text{ a.s. }\,,$$

*with $A_t' = c\sqrt{\frac{\mu}{L}}A_t$, with $A_0 = 1$.*

Following the lemma above, we get that:

$$0 \leq \mathbb{E}[\Phi(t, X_t, Y_t)] \leq \mathbb{E}[\Phi(0, X_0, Y_0)]\,.$$

We thus know that $A_t = e^{c\sqrt{\frac{\mu}{L}}}$, which implies that:

$$\mathbb{E}[A_t d_F(x_t, x^*)] \leq \mathbb{E}[\Phi(0, X_0, Y_0)]\,,$$

and we will obtain the conclusion of our theorem by expliciting all the constants in the following. We note that the expected number of activated edges between $[0, T]$ is by use of the Poisson Process $\int_0^T \mathrm{Tr}(\Lambda(t)\,dt$, and given the gradient fire at rate 1, the expected number of gradients computed is $nT$. $\qquad\square$

*Proof of Corollary 3.2.1.* We remind the SGD version of our Lemma:

**Lemma D.7.** *There exists some parameters* $\nu, \tilde{\eta}, \eta, \gamma, \tilde{\gamma}, \alpha, \tilde{\alpha}, \theta, \delta, \tilde{\delta}, \beta, \tilde{\beta}$ *and* $c > 0, C > 0$ *such that:*

$$\langle \nabla \Phi(t, X_t, Y_t), \begin{pmatrix} 1 \\ a_1(X_t, Y_t) \\ a_2(X_t, Y_t) \end{pmatrix} \rangle + \left( \Phi(t, X_t + b_1(X_t), Y_t) - \Phi(t, X_t, Y_t) \right)$$

$$+ \sum_{(i,j) \in \mathcal{E}(t)} \lambda_{ij}(t) \left( \Phi(t, X_t, Y_t + b_2^{ij}(Y_t)) - \Phi(t, X_t, Y_t) \right) \le C A_t \frac{1}{L} \ a.s. , \quad (20)$$

*with* $A_t' = c\sqrt{\frac{\mu}{L}} A_t$*, with* $A_0 = 1$.

The proof follows the same path, except that we have an extra term that writes for any $T > 0$:

$$\int_0^T \frac{A_t}{L} \le \frac{1}{c\sqrt{\mu L}} \quad (21)$$

which leads to the conclusion following an identical path. The constants will be explicited in the next Lemma. □

## D.4 PROOF OF THE LEMMA D.6 AND LEMMA D.7

We first state a couple of inequalities that we will combine to obtain a bound on our Lyapunov function.

**Proposition D.8.** *First:*

$$\phi_A \triangleq A_t(d_F(x^+, x^*) - d_F(x, x^*)) + \tilde{A}_t(\|\tilde{x}^+ - x^*\|^2 - \|\tilde{x} - x^*\|^2)$$
$$+ \eta A_t \langle \tilde{x} - x, \nabla F(x) - \nabla F(x^*) \rangle + 2\tilde{\eta} \tilde{A}_t \langle x - \tilde{x}, \tilde{x} - x^* \rangle \quad (22)$$

$$\le \|\nabla F(x) - \tilde{y}\|^2 \left( A_t \frac{L\gamma^2}{2} - A_t \gamma + \tilde{A}_t \tilde{\gamma}^2 \right)$$
$$+ A_t \gamma \langle \nabla F(x) - \tilde{y}, y^* - \tilde{y} \rangle + 2\tilde{\gamma} \tilde{A}_t \langle \tilde{y} - y^*, \tilde{x} - x^* \rangle \quad (23)$$
$$- 2\tilde{\gamma} \tilde{A}_t \left( d_F(\tilde{x}, x^*) + d_F(x^*, x) - d_F(\tilde{x}, x) \right)$$
$$- \eta A_t(d_F(\tilde{x}, x) + d_F(x, x^*) - d_F(\tilde{x}, x^*)) - \tilde{A}_t \tilde{\eta} \|\tilde{x} - x^*\|^2 + \tilde{A}_t \tilde{\eta} \|x - x^*\|^2$$

*Proof.* First, we have to use optimality conditions and smoothness:

$$d_F(x^+, x^*) - d_F(x, x^*) = d_F(x^+, x) - \langle x^+ - x, \nabla F(x^*) - \nabla F(x) \rangle \quad (24)$$

$$\le \frac{L}{2}\|x^+ - x\|^2 - \langle x^+ - x, \nabla F(x^*) - \nabla F(x) \rangle \quad (25)$$

$$= \frac{L\gamma^2}{2}\|\tilde{y} - \nabla F(x)\|^2 - \gamma\|\nabla F(x) - \tilde{y}\|^2$$
$$+ \gamma \langle \nabla F(x) - \tilde{y}, y^* - \tilde{y} \rangle \quad (26)$$

Next, we note that, again using optimality conditions:

$$\|\tilde{x}^+ - x^*\|^2 - \|\tilde{x}^+ - x^*\|^2 = 2\langle \tilde{x}^+ - \tilde{x}, \tilde{x} - x^* \rangle + \|\tilde{x}^+ - \tilde{x}\|^2 \quad (27)$$

$$= -2\tilde{\gamma}\langle \nabla F(x) - \tilde{y}, \tilde{x} - x^* \rangle + \tilde{\gamma}^2 \|\nabla F(x) - \tilde{y}\|^2 \quad (28)$$

$$= -2\tilde{\gamma}\langle \nabla F(x) - \nabla F(x^*), \tilde{x} - x^* \rangle$$
$$+ 2\tilde{\gamma}\langle \tilde{y} - y^*, \tilde{x} - x^* \rangle + \tilde{\gamma}^2 \|\nabla F(x) - \tilde{y}\|^2 \quad (29)$$

$$= -2\tilde{\gamma}(d_F(\tilde{x}, x^*) + d_F(x^*, x) - d_F(\tilde{x}, x))$$
$$+ 2\tilde{\gamma}\langle \tilde{y} - y^*, \tilde{x} - x^* \rangle + \tilde{\gamma}^2 \|\nabla F(x) - \tilde{y}\|^2 \quad (30)$$

Momentum in $x$ associated with the term $d_F(x, x^*)$ gives:

$$\eta \langle \tilde{x} - x, \nabla F(x) - \nabla F(x^*) \rangle = -\eta(d_F(\tilde{x}, x) + d_F(x, x^*) - d_F(\tilde{x}, x^*)) \quad (31)$$

and momentum in $\tilde{x}$ associated with $\|\tilde{x} - x^*\|^2$ leads to:

$$2\tilde{\eta}\langle x - \tilde{x}, \tilde{x} - x^*\rangle = -2\tilde{\eta}\|\tilde{x} - x^*\|^2 + 2\tilde{\eta}\langle x - x^*, \tilde{x} - x^*\rangle \leq -\tilde{\eta}\|\tilde{x} - x^*\|^2 + \tilde{\eta}\|x - x^*\|^2 \quad (32)$$

$\square$

**Corollary D.8.1.** *Under Assumption 3.4, we have:*

$$\tilde{\phi}_A \triangleq \mathbb{E}_\xi[A_t(d_F(x^+, x^*) - d_F(x, x^*)) + \tilde{A}_t(\|\tilde{x}^+ - x^*\|^2 - \|\tilde{x} - x^*\|^2)$$
$$+ \eta A_t\langle \tilde{x} - x, \nabla F(x) - \nabla F(x^*)\rangle + 2\tilde{\eta}\tilde{A}_t\langle x - \tilde{x}, \tilde{x} - x^*\rangle] \quad (33)$$

$$\leq \phi_A + \sigma^2\left(A_t\frac{L\gamma^2}{2} - A_t\gamma + \tilde{A}_t\tilde{\gamma}\right) \quad (34)$$

*Proof.* Using the same computations and the Eq. 36, we next note that:

$$\mathbb{E}_\xi[\|\nabla F(x, \xi) - y\|^2] = \mathbb{E}_\xi[\|\nabla F(x, \xi)\|^2 - 2\langle \nabla F(x, \xi), y\rangle + \|y\|^2] \quad (35)$$
$$\leq \|\nabla F(x) - y\|^2 + \sigma^2 \quad (36)$$

$\square$

**Proposition D.9.** *Next, we show that if $\alpha B_t = \frac{\delta}{2}\tilde{B}_t$:*

$$\phi_B \triangleq B_t(\|y^+ - y^*\|^2 - \|y - y^*\|^2) + \tilde{B}_t(\|\tilde{y}^+ - y^*\|^2 - \|\tilde{y} - y^*\|^2)$$
$$+ 2\alpha B_t\langle y - y^*, \tilde{y} - y\rangle - 2\theta\tilde{B}_t\langle y + z + \nu\tilde{x}, \tilde{y} - y^*\rangle$$
$$+ 2\alpha C_t\langle \tilde{y} - y, z + y - y^* - z^*\rangle \quad (37)$$

$$\leq -\frac{\delta}{2}\tilde{B}_t\|\tilde{y} - y^*\|^2 - \frac{\delta}{2}\tilde{B}_t\|y - y^*\|^2 - 2\tilde{\delta}\tilde{B}_t\langle \nabla F(x) - \tilde{y}, y^* - \tilde{y}\rangle$$
$$+ \delta\tilde{B}_t\|\nabla F(x) - \nabla F(x^*)\|^2 + \left((\delta + \tilde{\delta})^2 - \delta\right)\tilde{B}_t\|\nabla F(x) - y\|^2$$
$$- 2\theta\tilde{B}_t\langle y + z - y^* - z^*, \tilde{y} - y^*\rangle - 2\theta\nu\tilde{B}_t\langle \tilde{x} - x^*, \tilde{y} - y^*\rangle$$
$$+ 2\alpha C_t\langle \tilde{y} - y, z + y - y^* - z^*\rangle \quad (38)$$

*Proof.* Using optimality conditions:

$$\|\tilde{y}^+ - y^*\|^2 - \|\tilde{y} - y^*\|^2 = 2\langle \tilde{y} - y^*, \tilde{y}^+ - \tilde{y}\rangle + \|\tilde{y}^+ - \tilde{y}\|^2 \quad (39)$$
$$= 2\delta\langle \nabla F(x) - \tilde{y}, \tilde{y} - y^*\rangle + 2\tilde{\delta}\langle \nabla F(x) - \tilde{y}, \tilde{y} - y^*\rangle$$
$$(\delta + \tilde{\delta})^2\|\nabla F(x) - \tilde{y}\|^2 \quad (40)$$
$$= -2\tilde{\delta}\langle \nabla F(x) - \tilde{y}, y^* - \tilde{y}\rangle$$
$$+ \delta\|\nabla F(x) - \nabla F(x^*)\|^2 - \delta\|\tilde{y} - y^*\|^2$$
$$\left((\delta + \tilde{\delta})^2 - \delta\right)\|\nabla F(x) - \tilde{y}\|^2 \quad (41)$$

The momentum in $\tilde{y}$ associated with the term $\|\tilde{y} - y^*\|^2$ gives:

$$-2\theta\tilde{B}_t\langle y + z + \nu\tilde{x}, \tilde{y} - y^*\rangle = -2\theta\tilde{B}_t\langle y + z - y^* - z^*, \tilde{y} - y^*\rangle$$
$$- 2\theta\nu\tilde{B}_t\langle \tilde{x} - x^*, \tilde{y} - y^*\rangle \quad (42)$$

The momentum in $y$ associated with the term $\|y - y^*\|^2$ gives:

$$2\alpha B_t\langle \tilde{y} - y, y - y^*\rangle = -\alpha B_t\|y - y^*\|^2 - \alpha B_t\|\tilde{y} - y\|^2 + \alpha B_t\|\tilde{y} - y^*\|^2 \quad (43)$$

and the one associated with $\|y + z - y^* - z^*\|^2$:

$$2\alpha C_t\langle \tilde{y} - y, z + y - y^* - z^*\rangle \quad (44)$$

$\square$

**Corollary D.9.1.** *Under Assumption 3.4, we have:*

$$\tilde{\phi}_B \triangleq \mathbb{E}_\xi[\tilde{B}_t(\|y^+ - y^*\|^2 - \|y - y^*\|^2) + \tilde{B}_t(\|\tilde{y}^+ - y^*\|^2 - \|\tilde{y} - y^*\|^2)$$

$$+ 2\alpha B_t\langle y - y^*, \tilde{y} - y\rangle - 2\theta\tilde{B}_t\langle y + z + \nu\tilde{x}, \tilde{y} - y^*\rangle] \tag{45}$$

$$\leq \phi_B + \sigma^2((\delta^2 + (\delta + \tilde{\delta})^2)\tilde{B}_t) \tag{46}$$

*Proof.* Exactly as above. $\square$

**Proposition D.10.** *Finally, assuming $\theta\tilde{B}_t = \tilde{\beta}\tilde{C}_t = \alpha C_t$, letting $1 \geq \tilde{\tau} > 0$, $z_{ij}^+ = \beta(\mathbf{e}_i - \mathbf{e}_j)(\mathbf{e}_i - \mathbf{e}_j)^\mathsf{T}(y + z)$ and $\tilde{z}_{ij}^+ = \tilde{\beta}(\mathbf{e}_i - \mathbf{e}_j)(\mathbf{e}_i - \mathbf{e}_j)^\mathsf{T}(y + z)$, then:*

$$\phi_C + \phi_D - 2\theta\tilde{B}_t\langle y + z - y^* - z^*, \tilde{y} - y^*\rangle \triangleq$$

$$\sum_{ij}\lambda_{ij}(t)C_t\Big(\|y + z_{ij}^+ - y^* - z^*\|^2 - \|y + z - y^* - z^*\|^2\Big)$$

$$+ \sum_{ij}\lambda_{ij}(t)\tilde{C}_t\Big(\|\tilde{z}_{ij}^+ - z^*\|^2 - \|\tilde{z} - z^*\|^2\Big) + 2\tilde{\alpha}\tilde{C}_t\langle z - \tilde{z}, \tilde{z} - z^*\rangle_{\mathbf{\Lambda}(t)^+} \tag{47}$$

$$+ 2\alpha C_t\langle \tilde{z} + \tilde{y} - z^* - y^*, z + y - y^* - z^*\rangle - 2\theta\tilde{B}_t\langle y + z - y^* - z^*, \tilde{y} - y^*\rangle$$

$$\leq -2\tilde{\beta}\tilde{C}_t\langle \tilde{z} - z^*, \pi(y + z)\rangle + \tilde{\beta}^2\chi_2^*\tilde{C}_t\sum_{(i,j)\in\mathcal{E}(t)}\lambda_{ij}(t)\|(\mathbf{e}_i - \mathbf{e}_j)(\mathbf{e}_i - \mathbf{e}_j)^\mathsf{T}(y + z)\|^2$$

$$- \frac{\beta}{\chi_1^*}C_t\|\pi(y + z)\|^2 + \beta(\beta - 1)C_t\sum_{(i,j)\in\mathcal{E}(t)}\lambda_{ij}(t)\|(\mathbf{e}_i - \mathbf{e}_j)(\mathbf{e}_i - \mathbf{e}_j)^\mathsf{T}(y + z)\|^2$$

$$- \alpha C_t\|y + z - y^* - z^*\|^2 + \tilde{\alpha}\chi_1^*\tilde{C}_t\|z - z^*\|^2 - \tilde{\alpha}\tilde{C}_t\|\tilde{z} - z^*\|_{\mathbf{\Lambda}(t)^+}^2$$

$$- \tilde{\tau}\frac{1}{2}\tilde{\beta}\frac{\nu}{L}\tilde{C}_t\|z - z^*\|^2 + \tilde{\tau}\frac{\nu}{L}\frac{2\alpha\theta}{\delta}B_t\|y - y^*\|^2 \tag{48}$$

*Proof.* Having in mind that $\pi(y^* + z^*) = 0$ and $\mathbf{\Lambda}(t)^+\mathbf{\Lambda}(t) = \pi$, we get, using Lemma D.1 and Lemma D.3 on the inequality 52:

$$\Delta_{\tilde{z}} \triangleq \sum_{(i,j)\in\mathcal{E}(t)}\lambda_{ij}(t)\big(\|\tilde{z}_{ij}^+ - z^*\|_{\mathbf{\Lambda}(t)^+}^2 - \|\tilde{z} - z^*\|_{\mathbf{\Lambda}(t)^+}^2\big) \tag{49}$$

$$= \sum_{(i,j)\in\mathcal{E}(t)}\lambda_{ij}(t)2\langle \tilde{z} - z^*, \tilde{z}_{ij}^+ - \tilde{z}\rangle_{\mathbf{\Lambda}(t)^+} + \|\tilde{z}_{ij}^+ - \tilde{z}\|_{\mathbf{\Lambda}(t)^+}^2 \tag{50}$$

$$= -2\tilde{\beta}\sum_{(i,j)\in\mathcal{E}(t)}\lambda_{ij}(t)\langle \tilde{z} - z^*, (\mathbf{e}_i - \mathbf{e}_j)(\mathbf{e}_i - \mathbf{e}_j)^\mathsf{T}(y + z - y^* - z^*)\rangle_{\mathbf{\Lambda}(t)^+}$$

$$+ \sum_{(i,j)\in\mathcal{E}(t)}\lambda_{ij}(t)\tilde{\beta}^2\|(\mathbf{e}_i - \mathbf{e}_j)(\mathbf{e}_i - \mathbf{e}_j)^\mathsf{T}(y + z)\|_{\mathbf{\Lambda}(t)^+}^2 \tag{51}$$

$$\leq -2\tilde{\beta}\langle \tilde{z} - z^*, \mathbf{\Lambda}(t)^+\mathbf{\Lambda}(t)(y + z)\rangle$$

$$+ \chi_2^*\tilde{\beta}^2\sum_{(i,j)\in\mathcal{E}(t)}\lambda_{ij}(t)\|(\mathbf{e}_i - \mathbf{e}_j)(\mathbf{e}_i - \mathbf{e}_j)^\mathsf{T}(y + z)\|^2 \tag{52}$$

$$= -2\tilde{\beta}\langle \tilde{z} - z^*, \pi(y + z)\rangle + \chi_2^*\tilde{\beta}^2\sum_{(i,j)\in\mathcal{E}(t)}\lambda_{ij}(t)\|(\mathbf{e}_i - \mathbf{e}_j)(\mathbf{e}_i - \mathbf{e}_j)^\mathsf{T}(y + z)\|^2 \tag{53}$$

We also have, as $y^+ = y$ and using Lemma D.2:

$$\Delta_z \triangleq \sum_{(i,j)\in\mathcal{E}(t)} \lambda_{ij}(t)(\|y^+ + z_{ij}^+ - y^* - z^*\|^2 - \|y + z - y^* - z^*\|^2) \tag{54}$$

$$= 2 \sum_{(i,j)\in\mathcal{E}(t)} \lambda_{ij}(t)\langle y + z_{ij}^+ - y - z, y + z - y^* - z^*\rangle$$

$$+ \sum_{(i,j)\in\mathcal{E}(t)} \lambda_{ij}(t)\|y + z_{ij}^+ - y - z\|^2 \tag{55}$$

$$= -2 \sum_{(i,j)\in\mathcal{E}(t)} \beta\lambda_{ij}(t)\langle (\mathbf{e}_i - \mathbf{e}_j)(\mathbf{e}_i - \mathbf{e}_j)^\mathsf{T}(y + z), y + z - y^* - z^*\rangle$$

$$+ \sum_{(i,j)\in\mathcal{E}(t)} \beta^2\lambda_{ij}(t)\|(\mathbf{e}_i - \mathbf{e}_j)(\mathbf{e}_i - \mathbf{e}_j)^\mathsf{T}(y + z)\|^2 \tag{56}$$

$$= \sum_{(i,j)\in\mathcal{E}(t)} \lambda_{ij}(t)\Big( -\beta\|(\mathbf{e}_i - \mathbf{e}_j)(\mathbf{e}_i - \mathbf{e}_j)^\mathsf{T}(y + z)\|^2 - \beta\|\pi(y + z)\|^2$$

$$+ \beta\|(\mathbf{e}_i - \mathbf{e}_j)(\mathbf{e}_i - \mathbf{e}_j)^\mathsf{T}(y + z) - \pi(y + z)\|^2$$

$$+ \beta^2\|(\mathbf{e}_i - \mathbf{e}_j)(\mathbf{e}_i - \mathbf{e}_j)^\mathsf{T}(y + z)\|^2 \Big) \tag{57}$$

$$\leq -\frac{\beta}{\chi_1^*}\|\pi(y + z)\|^2 + \beta(\beta - 1) \sum_{(i,j)\in\mathcal{E}(t)} \lambda_{ij}(t)\|(\mathbf{e}_i - \mathbf{e}_j)(\mathbf{e}_i - \mathbf{e}_j)^\mathsf{T}(y + z)\|^2 \tag{58}$$

The momentum in $\tilde{z}$ associated with $\|\tilde{z} - z^*\|^2_{\boldsymbol{\Lambda}(t)^+}$ gives:

$$2\tilde{\alpha}\tilde{C}_t\langle z - \tilde{z}, \tilde{z} - z^*\rangle_{\boldsymbol{\Lambda}(t)^+} \leq \tilde{\alpha}\chi_1^*\tilde{C}_t\|z - z^*\|^2 - \tilde{\alpha}\tilde{C}_t\|\tilde{z} - z^*\|^2_{\boldsymbol{\Lambda}(t)^+} \tag{59}$$

And the one in $z$ associated with $\|y + z - y^* - z^*\|^2$ gives:

$$2\alpha C_t\langle \tilde{z} - z, z + y - y^* - z^*\rangle \tag{60}$$

Then, assuming that $\theta\tilde{B}_t = \tilde{\beta}\tilde{C}_t = \alpha C_t$, we have:

$$2\alpha C_t\langle \tilde{y} - y, z + y - y^* - z^*\rangle - 2\tilde{\beta}\tilde{C}_t\langle \tilde{z} - z^*, y + z - y^* - z^*\rangle$$

$$- 2\theta\tilde{B}_t\langle y + z - y^* - z^*, \tilde{y} - y^*\rangle + 2\alpha C_t\langle \tilde{z} - z, z + y - y^* - z^*\rangle \tag{61}$$

$$= -2\alpha C_t\|y + z - y^* - z^*\|^2 \tag{62}$$

At this stage, we split the negative term 62 in two halves, upper-bounding one of the halves by remembering that $\frac{\nu}{L} \leq 1$ and introducing $1 \geq \tilde{\tau} > 0$:

$$-\alpha C_t\|y + z - y^* - z^*\|^2 \leq -\tilde{\tau}\frac{\nu}{L}\alpha C_t\|y + z - y^* - z^*\|^2 \tag{63}$$

$$= -\tilde{\tau}\tilde{\beta}\frac{\nu}{L}\tilde{C}_t\|y + z - y^* - z^*\|^2 \tag{64}$$

$$\leq -\tilde{\tau}\frac{1}{2}\tilde{\beta}\frac{\nu}{L}\tilde{C}_t\|z - z^*\|^2 + \tilde{\tau}\tilde{\beta}\frac{\nu}{L}\tilde{C}_t\|y - y^*\|^2 \tag{65}$$

$$= -\tilde{\tau}\frac{1}{2}\tilde{\beta}\frac{\nu}{L}\tilde{C}_t\|z - z^*\|^2 + \tilde{\tau}\frac{\nu}{L}\frac{2\alpha\theta}{\delta}B_t\|y - y^*\|^2 \tag{66}$$

$$\square$$

Keeping in mind that $\theta\tilde{B}_t = \tilde{\beta}\tilde{C}_t = \alpha C_t$ and $\frac{\delta}{2}\tilde{B}_t = \alpha B_t$, we put everything together. Defining $\Psi = \phi_A + \phi_B + \phi_C + \phi_D$, we have:

$$\Psi \leq \|\nabla F(x) - \tilde{y}\|^2 \left( A_t \frac{L\gamma^2}{2} - A_t\gamma + \tilde{A}_t\tilde{\gamma}^2 + \left((\delta + \tilde{\delta})^2 - \delta\right)\tilde{B}_t \right) \tag{67}$$

$$+ \|\tilde{z} - z^*\|_{\mathbf{\Lambda}(t)^+}^2 \left( -\tilde{\alpha}\tilde{C}_t + \tilde{C}_t' \right) \tag{68}$$

$$+ \|\tilde{y} - y^*\|^2 (\tilde{B}_t' - \frac{\delta}{2}\tilde{B}_t) \tag{69}$$

$$+ \|x - x^*\|^2 (\tilde{A}_t\tilde{\eta} - \tilde{A}_t\frac{\nu\tilde{\gamma}}{2}) \tag{70}$$

$$+ \|\tilde{x} - x^*\|^2 (\tilde{A}_t' - \tilde{A}_t\tilde{\eta}) \tag{71}$$

$$+ \|\nabla F(x) - \nabla F(x^*)\|^2 (\delta\tilde{B}_t - \frac{\tilde{\gamma}}{2L}\tilde{A}_t) \tag{72}$$

$$+ \|\pi(y + z) - \pi(y^* + z^*)\|^2 (-\frac{\beta}{\chi_1^*}C_t) \tag{73}$$

$$+ \sum_{(i,j)\in\mathcal{E}(t)} \lambda_{ij}(t)\|(\mathbf{e}_i - \mathbf{e}_j)(\mathbf{e}_i - \mathbf{e}_j)^\mathsf{T}(y + z)\|^2 \left( \chi_2^*\tilde{\beta}^2\tilde{C}_t + \beta(\beta - 1)C_t \right) \tag{74}$$

$$+ \|z - z^*\|^2 (\chi_1^*\tilde{\alpha} - \tilde{\tau}\frac{1}{2}\tilde{\beta}\frac{\nu}{L})\tilde{C}_t \tag{75}$$

$$+ \|y - y^*\|(B_t' - (1 - \tilde{\tau}\frac{\nu}{L}\frac{2\theta}{\delta})\alpha B_t) \tag{76}$$

$$+ \|y + z - y^* - z^*\|^2 (C_t' - \alpha C_t) \tag{77}$$

$$+ d_F(x, x^*)(A_t' - \eta A_t) \tag{78}$$

$$+ d_F(\tilde{x}, x)(-A_t\eta + 2\tilde{\gamma}\tilde{A}_t) \tag{79}$$

$$+ d_F(\tilde{x}, x^*)(A_t\eta - 2\tilde{\gamma}\tilde{A}_t) \tag{80}$$

$$+ \langle\nabla F(x) - \tilde{y}, y^* - \tilde{y}\rangle(-2\tilde{\delta}\tilde{B}_t + \gamma A_t) \tag{81}$$

$$+ \langle\tilde{y} - y^*, \tilde{x} - x^*\rangle \left( 2\tilde{\gamma}\tilde{A}_t - 2\theta\nu\tilde{B}_t \right) \tag{82}$$

**Resolution GD**

*Proof of Lemma D.6.* Our goal is to put to zero all of the terms appearing next to scalar products and make the factors of positive quantities (norms or divergences) less or equal to zero. Given our relations, we guess that each exponential has the same rate. Thus, with $\tau > 0$, we fix $\frac{\delta}{2} = \tilde{\eta} = \eta = \tilde{\alpha} = \tau\sqrt{\frac{\nu}{L}}$, which leads to $\tilde{\gamma} = \frac{2\tau}{\sqrt{\nu L}}$ using Eq. 70. Also, from Eq. 80:

$$4\tilde{A}_t = \nu A_t.$$

Next, from Eq. 72 and Eq. 82, it's necessary that:

$$2L\delta = \theta\nu \,,$$

thus $\theta = 4\tau\sqrt{\frac{L}{\nu}}$. From Eq. 82, we get:

$$\tilde{A}_t = 2L\nu\tilde{B}_t.$$

Combining this previous equation with Eq. 81, as $4\tilde{A}_t = \nu A_t$, we have $\tilde{\delta} = 4L\gamma$. Next, Eq. 67 gives, with the equations above:

$$
\begin{aligned}
A_t(\frac{L\gamma^2}{2} - \gamma) + \tilde{A}_t\tilde{\gamma}^2 + \left((\delta + \tilde{\delta})^2 - \delta\right)\tilde{B}_t &= A_t\frac{L\gamma^2}{2} - A_t\gamma + \frac{\nu}{4}\tilde{\gamma}^2 A_t \\
&\quad + \left(\delta^2 + \tilde{\delta}^2 + \delta\right)\frac{A_t}{8L} \\
&= A_t\left(\frac{L\gamma^2}{2} - \gamma + \frac{\nu}{4}\frac{4\tau^2}{\nu L}\right) \\
&\quad + A_t(2\tau\sqrt{\frac{\nu}{L}} + 4\tau^2\frac{\nu}{L} + 16L^2\gamma^2)\frac{1}{8L} \\
&\le A_t(\gamma^2\frac{5}{2}L - \gamma + \frac{5}{4}\frac{\tau^2}{L} + \frac{\sqrt{2}}{8}\frac{\tau}{L})
\end{aligned}
$$

We thus pick $\gamma = \frac{1}{4L}$ and $\tau = \frac{1}{8}$, so that $\tilde{\delta} = 1$. Via Eq. 76, we fix $\tilde{\tau} = \frac{1}{8} < 1$. With Eq. 75, we then get:

$$\tilde{\beta} = 2\chi_1^*\sqrt{\frac{L}{\nu}}$$

We also put $\alpha = 2\tau\sqrt{\frac{\nu}{L}}$ and only one last equation, Eq. 74, needs to be satisfied, for which we pick $\beta = \frac{1}{2}$:

$$\chi_2^*\tilde{\beta}^2\tilde{C}_t + \beta(\beta - 1)C_t = (\chi_2^*\tilde{\beta}\alpha - \frac{1}{4})C_t$$

This implies that $\chi_2^*\chi_1^* \le \frac{1}{2}$. Finally, it's clear that all the equations are satisfied if we consider $A_t, \tilde{A}_t, B_t, \tilde{B}_t, C_t, \tilde{C}_t$ as exponentials proportional to $e^{\tau\sqrt{\frac{\nu}{L}}}$. Let's pick $A_0 = 1$.

Now, we remark that:

$$\Phi(0, X_0, Y_0) = A_0 d_F(x_0, x^*) + \tilde{A}_0\|\tilde{x}_0 - x^*\|^2 + B_0\|y_0 - y^*\|^2 + \tilde{B}_0\|\tilde{y}_0 - y^*\|^2 + C_0\|z_0 + y_0 - z^* - y^*\|^2 + \tilde{C}_0\|\tilde{z}_0 - z^*\|_{\Lambda(t)} +$$

If $\tilde{x}_0 = x_0$, $y_0 = \tilde{y}_0 = \nabla f(x_0)$ and $z_0 = \tilde{z}_0 = -\pi\nabla f(x_0)$, then, given the linear relation between $A_t, \tilde{A}_t, B_t, \tilde{B}_t, C_t, \tilde{C}_t$, the $L$ smoothness and the fact $\pi$ is an orthogonal projection, we get:

$$\Phi(0, X_0, Y_0) \le d_F(x_0, x^*) + \frac{\nu}{4}\|x_0 - x^*\|^2 + \frac{1}{8}d_F(x_0, x^*) + \frac{1}{16}d_F(x_0, x^*) + \frac{1}{4}d_F(x_0, x^*) \tag{83}$$

$$+ \frac{1}{32}\frac{\chi_1(0)}{\chi_1^*}\frac{\nu}{L}d_F(x_0, x^*). \tag{84}$$

Now, we use that $\nu = \frac{\mu}{2} \le L$ and as $\chi_1(0) \le \chi_1^*$, we get:

$$\Phi(0, X_0, Y_0) \le 2d_F(x_0, x^*) + \frac{\mu}{8}\|x_0 - x^*\|^2.$$

Given that $d_F(x, x^*) = f(x) - f(x^*)$, this implies in particular that:

$$\mathbb{E}[f(x_t)] - f(x^*) \le \left(2d_F(x_0, x^*) + \frac{\mu}{8}\|x_0 - x^*\|^2\right)e^{-\frac{t}{8}\sqrt{\frac{\nu}{L}}}$$

$\square$

**Resolution SGD**

*Proof of Lemma D.7.* All the previous computations hold, except that the term in front of $\sigma^2$ is given by:

$$(\delta^2 + (\delta + \tilde{\delta})^2)\tilde{B}_t + (A_t\frac{L\gamma^2}{2} - A_t\gamma + \tilde{A}_t\tilde{\gamma}) = (\delta^2 + (\delta + \tilde{\delta})^2)\frac{A_t}{8L}$$

$$+ (A_t\frac{L\gamma^2}{2} - A_t\gamma + \nu\frac{A_t}{4}\tilde{\gamma}) \qquad (85)$$

$$\leq \frac{5}{8L}A_t \qquad (86)$$

Thus, we obtain, by integration of the potential that:

$$\Phi(t, X_t, Y_t) \leq \Phi(0, X_0, Y_0) + \sigma^2\int_0^t \frac{5}{8L}A_u\,du \qquad (87)$$

Now with $A_t, \tilde{A}_t, B_t, \tilde{B}_t, C_t, \tilde{C}_t$ as above and all the constants as above, we get the result, since: $\int_0^t \frac{5}{8L}A_u\,du \leq \frac{5}{\sqrt{L}\mu}$.

$\square$

## E  PHYSICAL INTERPRETATION

To gain more insight on the condition $2\chi_1^*[\Lambda]\chi_2^*[\Lambda] \leq 1$, we can write $\Lambda(t)$ as the product of two more interpretable quantities:

$$\Lambda(t) = \underbrace{\sum_{(ij)\in\mathcal{E}(t)} \lambda_{ij}(t)}_{\triangleq\lambda(t)} \underbrace{\frac{2\Lambda(t)}{\text{Tr }\Lambda(t)}}_{\triangleq\tilde{\Lambda}(t)}. \qquad (88)$$

In this setting, $\lambda(t)$ is the instantaneous expected rate of communication over the whole graph at time $t$, while $\tilde{\Lambda}(t)$ can be interpreted as the Laplacian of $\mathcal{E}(t)$ with each edge weighted with its probability of having spiked at this instant given an edge fired at time $t$.

Being normalized, $\tilde{\Lambda}(t)$ only contains the information about the graph's connectivity at time $t$ while $\lambda(t)$ is the global rate of communication. We have:

$$\chi_1[\Lambda(t)] = \frac{\chi_1[\tilde{\Lambda}(t)]}{\lambda(t)} \quad ; \quad \chi_2[\Lambda(t)] = \frac{\chi_2[\tilde{\Lambda}(t)]}{\lambda(t)}. \qquad (89)$$

If we make the following assumptions,

**Assumption E.1.** *There is a $\lambda^* > 0$ such that, at all time $t$, $\lambda(t) \geq \lambda^*$.*

**Assumption E.2.** *There are $\tilde{\chi}_1^* > 0$, $\tilde{\chi}_2^* > 0$ such that, for all $t$, $\chi_1[\tilde{\Lambda}(t)] \leq \tilde{\chi}_1^*$ and $\chi_2[\tilde{\Lambda}(t)] \leq \tilde{\chi}_2^*$.*

meaning we assume bounds on the worst rate of communication and on the worst graph connectivity, we immediately have $\chi_1[\Lambda(t)] \leq \frac{\tilde{\chi}_1^*}{\lambda^*}$ and $\chi_2[\Lambda(t)] \leq \frac{\tilde{\chi}_2^*}{\lambda^*}$, leading to $\chi_1^* \leq \frac{\tilde{\chi}_1^*}{\lambda^*}$ and $\chi_2^* \leq \frac{\tilde{\chi}_2^*}{\lambda^*}$. Then, if the following condition on the worst rate of communication is met

$$\sqrt{2\tilde{\chi}_1^*\tilde{\chi}_2^*} \leq \lambda^*, \qquad (90)$$

meaning that the instantaneous global communication rate is always larger than some spectral quantity quantifying the graph's connectivity, it directly implies $2\chi_1^*[\Lambda]\chi_2^*[\Lambda] \leq 1$ and the convergence of our method.

## F  COMPARISON WITH OTHER WORKS

We now explain the results of Sec. 4.1.

## F.1 COMPARISON WITH ADOM+

Using the notations of Kovalev et al. (2021a), we know that gossip matrices satisfy, for $q \in \mathbb{N}$:

$$\|W(q)x - x\|^2 \leq (1 - \frac{1}{\chi})\|x\|^2 \,,$$

for some $\chi \geq 1$. It implies that:

$$\mathrm{sp}(W(q)) \subset [1 - \sqrt{1 - \frac{1}{\chi}}, 2] \,,$$

and for $\chi$ large enough, $1 - \sqrt{1 - \frac{1}{\chi}} \approx \frac{1}{2\chi}$. Consequently, up to a renormalization factor, we have $\chi_1^*[W] \approx 2\chi$ and:

$$\mathrm{Tr}(W(q)) \leq 2n \,.$$

## F.2 ACCELERATION OF THE CONTINUIZED FRAMEWORK

Under the notation of Even et al. (2021a), we note that, an additional simplification holds: $\theta'_{\mathrm{ARG}} = \theta_{\mathrm{ARG}}$. We remind that $\mathcal{L} = AA^\mathsf{T}$ and that $Ae_{vw} = \sqrt{P_{vw}}(e_v - e_w)$. Next, we note that by definition:

$$\frac{R_{vw}}{P_{vw}} \triangleq \frac{e_{vw}^\mathsf{T} A^+ A e_{vw}}{P_{vw}} \tag{91}$$

$$= \frac{e_{vw}^\mathsf{T} A^+ (e_v - e_w)}{\sqrt{P_{vw}}} \tag{92}$$

$$= \frac{(A^{+\mathsf{T}} e_{vw})^\mathsf{T}(e_v - e_w)}{\sqrt{P_{vw}}} \tag{93}$$

$$= \frac{((AA^\mathsf{T})^{+\mathsf{T}} A e_{vw})^\mathsf{T}(e_v - e_w)}{\sqrt{P_{vw}}} \tag{94}$$

$$= (e_v - e_w)^\mathsf{T} \mathcal{L}^+ (e_v - e_w) \,, \tag{95}$$

so we can directly relate their bounds to ours. Next, as $\mathcal{L}\mathcal{L}^+ = \mathbf{I} - \pi$, we observe that:

$$n - 1 = \mathrm{Tr}(\mathcal{L}\mathcal{L}^+) \tag{96}$$

$$= \sum_{(ij) \in \mathcal{E}} P_{ij}(e_i - e_j)^\mathsf{T} \mathcal{L}^+ (e_i - e_j) \tag{97}$$

$$\leq 2\chi_2[\mathcal{L}] \sum_{(ij) \in \mathcal{E}} P_{ij} \tag{98}$$

$$= \chi_2[\mathcal{L}]\mathrm{Tr}\mathcal{L} \,, \tag{99}$$

which, together with the fact that $\chi_1[\mathcal{L}] \geq \frac{n-1}{\mathrm{Tr}\mathcal{L}}$ (see Lemma 3.1), leads to:

$$n - 1 \leq 2\sqrt{\chi_1^*[\mathcal{L}]\chi_2^*[\mathcal{L}]}$$

in the setting of Even et al. (2021a) where $\mathrm{Tr}\,\mathcal{L} = 2$.

## F.3 COMPARISON WITH METHODS THAT USE THE SPECTRAL GAP

We note that: $\sqrt{\rho^*}|\mathcal{E}(t)| = \sqrt{\chi_1}\sqrt{\|\Lambda(t)\|}|\mathcal{E}(t)|$. Further, we assume $\frac{\sup_{(i,j) \in \mathcal{E}(t)} \lambda_{ij}(t)}{\inf_{(i,j) \in \mathcal{E}(t)} \lambda_{ij}(t)} = \mathcal{O}(1)$ as $n$ grows. This condition can be understood as a way to prevent degenerated behaviors in the network's connectivity: the worst case communication rate should always be greater than some fraction of the largest rate, with a fraction value not growing with the network's size. This condition is always met if we assume there are both a lower and upper bound on the communication rate of the channels linking the nodes, which seems reasonable in a physical setting. Then, using Lemma 3.1, recalling

that $\mathrm{Tr}(\Lambda(t)) = 2\sum_{(ij)\in\mathcal{E}(t)} \lambda_{ij}(t) \leq 2|\mathcal{E}(t)|\sup_{(i,j)\in\mathcal{E}(t)} \lambda_{ij}(t)$ and $\mathrm{Tr}(\Lambda(t)) \leq (n-1)\|\Lambda(t)\|$, we obtain:

$$\sqrt{\chi_2}\mathrm{Tr}(\Lambda(t)) \leq \frac{1}{\sqrt{2\inf_{(i,j)\in\mathcal{E}(t)} \lambda_{ij}(t)}}\sqrt{\mathrm{Tr}\Lambda(t)}\sqrt{\mathrm{Tr}\Lambda(t)} \tag{100}$$

$$\sqrt{\chi_2}\mathrm{Tr}(\Lambda(t)) \leq \frac{1}{\sqrt{2\inf_{(i,j)\in\mathcal{E}(t)} \lambda_{ij}(t)}}\sqrt{2|\mathcal{E}(t)|\sup_{(i,j)\in\mathcal{E}(t)} \lambda_{ij}(t)}\sqrt{\mathrm{Tr}\Lambda(t)} \tag{101}$$

$$\leq \sqrt{\frac{\sup_{(i,j)\in\mathcal{E}(t)} \lambda_{ij}(t)}{\inf_{(i,j)\in\mathcal{E}(t)} \lambda_{ij}(t)}}\sqrt{|\mathcal{E}(t)|}\sqrt{(n-1)\|\Lambda(t)\|} \tag{102}$$

$$\leq \sqrt{\frac{\sup_{(i,j)\in\mathcal{E}(t)} \lambda_{ij}(t)}{\inf_{(i,j)\in\mathcal{E}(t)} \lambda_{ij}(t)}}|\mathcal{E}(t)|\sqrt{\|\Lambda(t)\|} \tag{103}$$

## G  FURTHER EXPERIMENTS

In this section, we present additional numerical results comparing our method DADAO to ADOM+ (Kovalev et al., 2021a) in the time-varying setting and report our results using SGD.

### G.1  TIME-VARYING SETTING

In this section, we study the effect of the parameter $\chi_1^*$ on the convergence speed of ADOM+ (Kovalev et al., 2021a) and DADAO by varying it between $\chi_1^* \in \{3, 33, 180, 233\}$ for random geometric graphs of size $n = 20$ on the decentralized linear regression task with time-varying topology. To visualize the difference in connectivity these changes in $\chi_1^*$ represent, we plot four graphs of the said types with varying values of $\chi_1^*$ in Fig. 3. In Fig. 4, we show the different convergence speeds it entails.

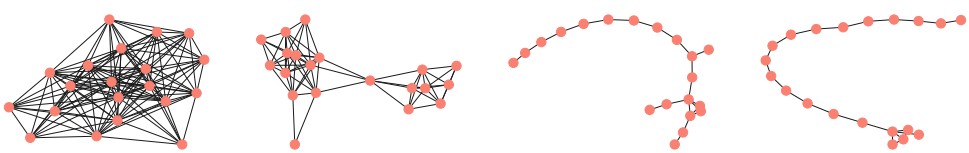

Figure 3: Examples of random geometric graphs of size $n = 20$ with $\chi_1^*$ taking values in, from left to right, $\chi_1^* \in \{3, 33, 180, 233\}$.

As expected, we observe in Fig. 4 that varying $\chi_1^*$ does not affect the number of gradient computations of both ADOM+ M.-C and DADAO, but the smaller the $\chi_1^*$, the better the slope for ADOM+ in terms of gradient steps. We also confirm for all three methods that the smaller $\chi_1^*$, the less communication is needed to reach an $\epsilon$-precision.

### G.2  STOCHASTIC GRADIENT DESCENT WITH DADAO

In the SGD setting, we randomly sample a mini-batch of size $B$ data points on each worker and compute the losses and stochastic gradients $\nabla f_i(x_i, \xi)$ w.r.t. these samples. To study the effect of the quadratic error $\sigma^2$ of our gradients on the resulting biases of our parameters, we fix both the data (for linear regression) and the communication network (graph star of size $n = 20$) and try different values of $B$. To monitor our results, we plot the mean distance to $x^*$ of the running average over time of our local parameters. Then, taking the notations introduced in Sec. 4.2, we can write:

$$\frac{1}{n}\sum_{i=1}^{n}\left\|\frac{1}{k_i}\sum_{j=1}^{k_i} x_j^{(i)} - x^*\right\|^2,$$

where $k_i$ designates a local event counter. We report our results in Fig. 5.

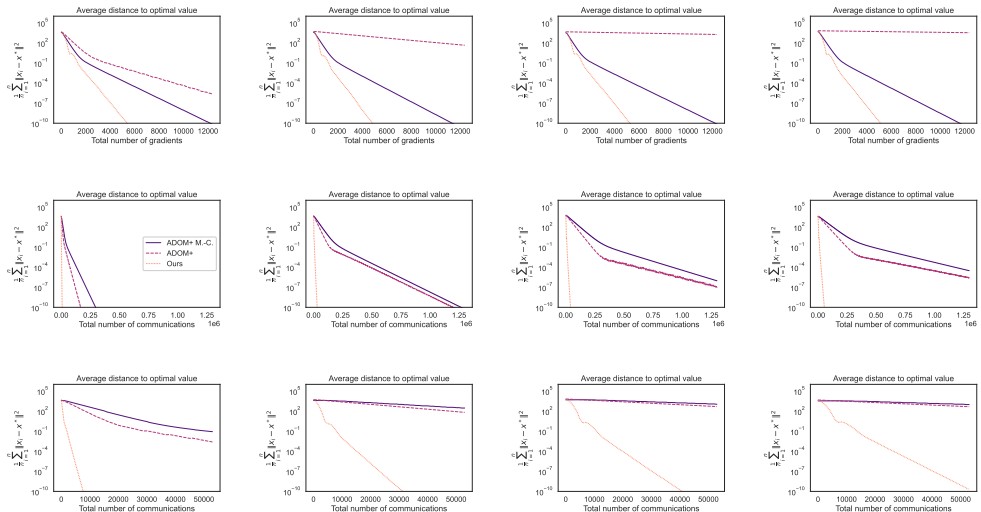

Figure 4: Comparison between ADOM+ (Kovalev et al., 2021a) and DADAO, using the same data for linear regression on $n = 20$ workers and the same sequence of random connected graphs with varying topology and $\chi_1^*$ taking values in, from the left to the right column, $\chi_1^* \in \{3, 33, 180, 233\}$.

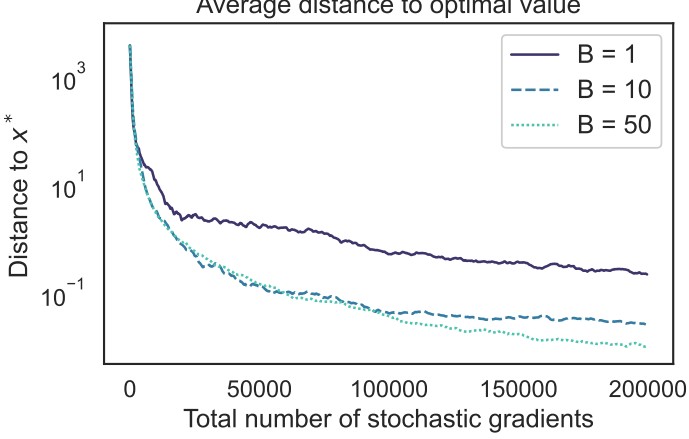

Figure 5: Effect of the batch size $B$ on the convergence of our method DADAO. Recall that the full batch size $m$ equals $100$.

We confirm that the less variance on our stochastic gradients, the less our estimates $\frac{1}{k_i} \sum_{j=1}^{k_i} x_j^{(i)}$ are biased.

### G.3 COMPARISON BETWEEN DADAO AND MSDA ON THE STAR GRAPH

For star graphs of size $n \in \{10, 20, 70, 200, 300, 1000, 2000\}$, we ran DADAO and MSDA on the task of distributed linear regression. We considered the evolution of the average distance to the optimal with the number of gradient steps and commmunication steps in log scale for each run, and computed the slope of each line. For each graph size, we report in Fig. 6 the rate between the slope for MSDA and the slope for DADAO. We remark that the rate between the gradient complexities

Figure 6: Rate between the slopes of MSDA and DADAO for star graphs of size $n \in \{10, 20, 70, 200, 300, 1000, 2000\}$.

of DADAO and MSDA is indeed a $\mathcal{O}(1)$ (with a constant value of $\simeq 14$) while MSDA is indeed $\mathcal{O}(\sqrt{n})$ worse than DADAO for communications on the star graph, as stated in Tab. 2.

## H    PRACTICAL IMPLEMENTATION

In this section, we describe in more detail the implementation of our algorithm. As we did not physically execute our method on a compute network but carried it out on a single machine, all the asynchronous computations and communications had to be simulated. Thus, we will first discuss the method we followed to simulate our asynchronous framework before detailing the practical steps of our algorithm through a pseudo-code.

### H.1    SIMULATING THE POISSON POINT PROCESSES

To emulate the asynchronous setting, before running our algorithm, we generate 2 independent sequences of jump times at the graph's scale: one for the computations and one for the communications. As we considered independent P.P.Ps, the time increments follow a Poisson distribution. At the graph's scale, each node spiking at a rate of 1, the Poisson parameter for the gradient steps process is $n$. Following the experimental setting of the Continuized framework (Even et al., 2021a), we considered that all edges in $\mathcal{E}(t)$ had the same probability of spiking between $t$ and $t + dt$. Thus, given the sequence of graphs $\mathcal{E}(t)$ and $\mathcal{L}(t)$ their corresponding Laplacians, we computed the parameter $\lambda^*$ of the communication process as such:

$$\lambda^* = \sqrt{2 \sup_t \chi_1 \left[ \frac{\mathcal{L}(t)}{|\mathcal{E}(t)|} \right] \sup_t \chi_2 \left[ \frac{\mathcal{L}(t)}{|\mathcal{E}(t)|} \right]}. \tag{104}$$

Having generated the 2 sequences of spiking times at the graph's scale, we run our algorithm playing the events in order of appearance, attributing the *location* of the events by sampling uniformly one node if the event is a gradient step or sampling uniformly an edge in $\mathcal{E}(t)$ if it is a communication.

### H.2    PSEUDO CODE

We keep the notations introduced in Eq. 3 and recall the following constant values specified in Appendix D.4:

$$\eta = \tfrac{1}{8}\sqrt{\tfrac{\nu}{L}} \quad \gamma = \tfrac{1}{4L} \quad \delta = \tfrac{1}{4}\sqrt{\tfrac{\nu}{L}} \quad \alpha = \tfrac{1}{4}\sqrt{\tfrac{\nu}{L}} \quad \beta = \tfrac{1}{2} \qquad \theta = \tfrac{1}{2}\sqrt{\tfrac{L}{\nu}}$$
$$\tilde{\eta} = \tfrac{1}{8}\sqrt{\tfrac{\nu}{L}} \quad \tilde{\gamma} = \tfrac{1}{4\sqrt{\nu L}} \quad \tilde{\delta} = 1 \quad \tilde{\alpha} = \tfrac{1}{8}\sqrt{\tfrac{\nu}{L}} \quad \tilde{\beta} = 2\chi_1^*[\Lambda]\sqrt{\tfrac{L}{\nu}} \quad \nu = \tfrac{\mu}{2}$$

For the sake of completeness, we also specify the matrix $\mathcal{A}$ describing the linear ODE 6:

$$\mathcal{A} = \begin{pmatrix} -\eta & \eta & 0 & 0 & 0 & 0 \\ \tilde{\eta} & -\tilde{\eta} & 0 & 0 & 0 & 0 \\ 0 & 0 & -\alpha & \alpha & 0 & 0 \\ 0 & -\theta\nu & -\theta & 0 & -\theta & 0 \\ 0 & 0 & 0 & 0 & -\alpha & \alpha \\ 0 & 0 & 0 & 0 & \tilde{\alpha} & -\tilde{\alpha} \end{pmatrix} \tag{105}$$

As described in Appendix H.1, we call `PPPspikes` the process mentioned above, returning the ordered sequence of events and time of spikes of the two P.P.Ps. Then, we can write the pseudo-code of our implementation of the DADAO optimizer in Algorithm 2.

---

**Algorithm 2:** Pseudo-code of our implementation of DADAO on a single machine.

---

**Input:** On each machine $i \in \{1, ..., n\}$, an oracle able to evaluate $\nabla f_i$, Parameters
$\mu, L, \chi_1^*, t_{\max}, n, \lambda^*$.
The sequence of time-varying graphs $\mathcal{E}(t)$.

1 **Initialize** on each machine $i \in \{1, ..., n\}$:
2      Set $X^{(i)} = (x_i, \tilde{x}_i, \tilde{y}_i)$ and $Y^{(i)} = (y_i, z_i, \tilde{z}_i)$ to 0 ;
3      Set constants $\nu, \tilde{\eta}, \eta, \gamma, \alpha, \tilde{\alpha}, \theta, \delta, \tilde{\delta}, \beta, \tilde{\beta}$ using $\mu, L, \chi_1^*$;
4      Set $\mathcal{A}$;
5      $T^{(i)} \leftarrow 0$ ;
6 ListEvents, ListTimes $\leftarrow$ PPPspikes$(n, \lambda^*, t_{\max})$ ;
7 $n_{\text{events}} \leftarrow |\text{ListEvents}|$ ;
8 **for** $k \in [\![1, n_{\text{events}}]\!]$ **do**
9      **if** `ListEvents[k]` *is to take a gradient step* **then**
10          $i \sim \mathcal{U}([\![1, n]\!])$ ;
11          $\begin{pmatrix} X^{(i)} \\ Y^{(i)} \end{pmatrix} \leftarrow \exp\left((\text{ListTimes}[k] - T^{(i)})\mathcal{A}\right) \begin{pmatrix} X^{(i)} \\ Y^{(i)} \end{pmatrix}$ ;
12          $x_i \leftarrow x_i - \gamma\left(\nabla f_i(x_i) - \nu x_i - \tilde{y}_i\right)$;
13          $\tilde{x}_i \leftarrow \tilde{x}_i - \tilde{\gamma}\left(\nabla f_i(x_i) - \nu x_i - \tilde{y}_i\right)$;
14          $\tilde{y}_i \leftarrow \tilde{y}_i + (\delta + \tilde{\delta})\left(\nabla f_i(x_i) - \nu x_i - \tilde{y}_i\right)$;
15          $T^{(i)} \leftarrow \text{ListTimes}[k]$ ;
16      **else if** `ListEvents[k]` *is to take a communication step* **then**
17          $(i, j) \sim \mathcal{U}(\mathcal{E}(\text{ListTimes}[k]))$ ;
18          $\begin{pmatrix} X^{(i)} \\ Y^{(i)} \end{pmatrix} \leftarrow \exp\left((\text{ListTimes}[k] - T^{(i)})\mathcal{A}\right) \begin{pmatrix} X^{(i)} \\ Y^{(i)} \end{pmatrix}$ ;
19          $\begin{pmatrix} X^{(j)} \\ Y^{(j)} \end{pmatrix} \leftarrow \exp\left((\text{ListTimes}[k] - T^{(j)})\mathcal{A}\right) \begin{pmatrix} X^{(j)} \\ Y^{(j)} \end{pmatrix}$ ;
20          $m_{ij} \leftarrow (y_i + z_i - y_j - z_j)$;                // Message exchanged.
21          $z_i \leftarrow z_i - \beta m_{ij}$;
22          $\tilde{z}_i \leftarrow \tilde{z}_i - \tilde{\beta} m_{ij}$;
23          $z_j \leftarrow z_j + \beta m_{ij}$;
24          $\tilde{z}_j \leftarrow \tilde{z}_j + \tilde{\beta} m_{ij}$;
25          $T^{(i)} \leftarrow \text{ListTimes}[k]$;
26          $T^{(j)} \leftarrow \text{ListTimes}[k]$;
27 **return** $(x_i)_{1 \le i \le n}$, *the estimate of $x^*$ on each worker $i$.*

---

