# OpenReview forum: "DADAO: Decoupled Accelerated Decentralized Asynchronous Optimization"
_ICLR.cc/2023/Conference — Submitted to ICLR 2023_

### Official Review · Reviewer_c2yc · 2022-10-24

**Confidence:** 5
**Correctness:** 2
**Technical Novelty And Significance:** 2
**Empirical Novelty And Significance:** 2
**Recommendation:** 3

**Clarity, Quality, Novelty And Reproducibility:**

Clarity: Poor
Quality: Poor
Novelty: Neutral
Reproducibility: N/A

**Strength And Weaknesses:**

Strength:
1. Theoretical convergence rate is established.
2. Studying the convergence from the continuous perspective.


Weaknesses:
1. The convergence rate in the abstract is wrong.
2. Why is it necessary to study the convergence rate from a continuous perspective? Is there any benefit? For strongly convex problems, the traditional analysis can also establish the linear convergence rate. Moreover, in the appendix, I didn't find the authors use continuous tools to investigate convergence. Why do you introduce Eq.(3) and that in Section 3.3?
3. What's the meaning of $\chi_1$ and $\chi_2$? How do they relate to the spectral gap?
4. How does the communication latency affect the convergence rate?
5. How does the spectral gap affect the convergence rate?
6. Can the convergence rate achieve the linear speedup with respect to the number of devices?
7. The experiment is too simple. More complicated models and datasets should be used to evaluate the performance.
8. Asynchronous decentralized SGD has been studied before. But the authors missed some important literature, e.g., http://proceedings.mlr.press/v80/lian18a.html

**Summary Of The Paper:**

This paper studied asynchronous decentralized SGD for strongly convex problems. The authors provide theoretical analysis for the convergence rate and conduct experiments to verify its performance. But the writing is not very clear. It is not easy to follow.

**Summary Of The Review:**

This paper studied asynchronous decentralized SGD for strongly convex problems. The authors provide theoretical analysis for the convergence rate and conduct experiments to verify its performance. But the writing is not very clear. It is not easy to follow.

---

> ### Author Response · Authors · 2022-11-08
> **Author response to reviewer c2yc**
>
> We thank the reviewer for the comments about our paper.
>
> **Strength:** We are glad that the reviewer agrees that we have a theoretical convergence rate - but it is also good to observe that those rates are state-of-the-art!(see Table 1, Table 2) However, if the reviewer is aware of concurrent works which outperform DADAO in a convex setting, we would be glad to compare them. Furthermore, our algorithm is NOT continuous but continuized: it is a discrete algorithm with a continuous analysis (there is no ODE but rather an SDE!)
>
> **Weakness:**
>
> * The notation $f= \mathcal O (g)$ often means there is $C>0$ such that $|f|\leq C|g|$: we have now restated our rate with $\log \frac{1}{\epsilon}$ to avoid any possible confusions.
> * Thanks for pointing this out, but DADAO is not a continuous algorithm: DADAO is based on an SDE (eq 3) with Pointwise Poisson Process: it is thus, by nature, a discrete algorithm with continuous proof. Indeed, the entire proof can be found in Appendix D3 and clearly uses the Ito Lemma, which is characteristic of SDE analysis. Please note that we use the same framework for proof as the last Neurips 2021 best paper [1]; however, we substantially improved its rates as shown in Table 1 and explained in Section 4.1 !
> * Thank you for asking. As written in our main text, those constants measure the connectivity of a graph, and as stated in the main text Sec. 41 and Appendix F, $\sqrt{\chi_1\chi_2}$ leads to better rates than the spectral gap. See [2] for a reference on the effective resistance of a graph
> * Thanks. We have restated Theorem 3.2 to state the communication complexity explicit: it is simply the trace of our Laplacian matrix, as it corresponds to the expected number of edges that spike per unit of time. As for communication delays, we took the standard approach of not integrating them in our model, but taking them into account would indeed be of interest and possible with our formulation, so it is left for future work.
> * It does not, and this is what makes our methods interesting! Our rate only depends on $\sqrt{\chi_1\chi_2}$.
> * Given that methods like MSDA or ADOM+ achieve this linear rate and that our work outperforms MSDA and ADOM+, yes, we do!
> * Thank you for pointing out this issue: if the reviewer is aware of more complex benchmarks and smooth and strongly convex losses, we would be glad to try them with our source code! Indeed, we simply took the same benchmarks as the reference works to which we compared.
> * Thank you for pointing out this paper, but given that they do not derive any convergence rates in a convex setting, we prefer avoiding confusing readers.
>
>
> [1] Mathieu Even, Raphaël Berthier, Francis Bach, Nicolas Flammarion, Hadrien Hendrikx, Pierre Gaillard, Laurent Massoulié, and Adrien Taylor. A continuized view on nesterov acceleration for stochastic gradient descent and randomized gossip. In Advances in Neural Information Processing Systems, 2021.
>
> [2] Wendy Ellens, Floske M Spieksma, Piet Van Mieghem, Almerima Jamakovic, and Robert E Kooij. Effective graph resistance. In Linear algebra and its applications, 2011.

---

### Official Review · Reviewer_R4MY · 2022-10-24

**Confidence:** 3
**Clarity, Quality, Novelty And Reproducibility:** writing is not very clear to follow
**Correctness:** 3
**Technical Novelty And Significance:** 2
**Empirical Novelty And Significance:** Not applicable
**Recommendation:** 5

**Strength And Weaknesses:**

strength: the proposed approach is differ from the majority of works since it model the local gradient updates and gossip communication procedures with separate independent Poisson Point Processes, to decouple the computation and communication steps.

weakness:
- the experiment setting with decentralized linear and logistic regression tasks is too simple to validate the effectiveness of the approach.
- many importance factors of convergence rate are not clear, such as communication latency and linear speedup

**Summary Of The Paper:**

This work proposed a decentralized asynchronous stochastic first order algorithm to minimize a sum of L-smooth and µ-strongly convex functions distributed over a time-varying connectivity network of size n. The authors model the local gradient updates and gossip communication procedures with separate independent Poisson Point Processes, to decouple the computation and communication steps. The proposed method is differ from the majority of works since it does not use a multi-consensus inner loop nor other ad-hoc mechanisms such as Error Feedback, Gradient Tracking, or a Proximal operator.


**Summary Of The Review:**

The authors model the local gradient updates and gossip communication procedures with separate independent Poisson Point Processes, which differ from the majority of works. The experiment is too simple for verification. The convergence analysis is lacking some important factors.

---

> ### Author Response · Authors · 2022-11-08
> **Author response to reviewer R4MY**
>
> We thank the reviewer for the comments about our paper.
>
> **Strength:** Thanks! Our method outperforms all related work to which we compared. If the reviewer feels we miss some references, we would gladly compare them.
>
> **Weakness:**
> * _Experiments._ We followed the standard approach for experiments in the context of decentralized optimization of a sum of smooth and strongly convex functions (see, e.g. [1,2,3]). However, if the reviewer is aware of other standards, we would be glad to compare them! Note that our methods provably outperforms other works.
> * _Communication and convergence rate:_ Thanks for pointing out this issue! We restated our theorem to show the standard convergence rate for smooth and strongly convex functions. In addition, we added a reference in Theorem 3.2 to Appendix H2, which states the hyper-parameters used, and we made explicit all the constants of our work, as well as the complexity and communication rate. We hope this clarifies the doubts of the reviewer.
> * _Communication latency:_ Thanks for suggesting this addition. We did not integrate communication and computation delays in our model, as is standard in the literature (see references), but taking them into account would indeed be of interest and possible with our formulation, so it is left for future work.
>
> [1] Anastasia Koloskova, Nicolas Loizou, Sadra Boreiri, Martin Jaggi, Sebastian Stich. A Unified Theory of Decentralized SGD with Changing Topology and Local Updates. In Proceedings of the 37th International Conference on Machine Learning, 2020.
>
> [2] Huan Li and Zhouchen Lin. Accelerated gradient tracking over time-varying graphs for decentralized optimization. arXiv preprint arXiv:2104.02596, 2021.
>
> [3] Dmitry Kovalev, Elnur Gasanov, Alexander Gasnikov, and Peter Richtarik, Lower Bounds and Optimal Algorithms for Smooth and Strongly Convex Decentralized Optimization Over Time-Varying Networks, In Advances in Neural Information Processing Systems, 2021.

---

### Official Review · Reviewer_gkNG · 2022-10-25

**Confidence:** 3
**Correctness:** 4
**Technical Novelty And Significance:** 2
**Empirical Novelty And Significance:** 2
**Recommendation:** 5

**Clarity, Quality, Novelty And Reproducibility:**

The problem setting is clear but the writing is not very good because of typos and missing definitions. The work seems to be novel.





**Strength And Weaknesses:**

Strength:

1. The proposed algorithm achieves new performance guarantees in both communication edges and gradients.

Weaknesses:

1. Theorems 3.2 and Corollary 3.2.1 provide guarantees for the ODE system Eq.(3) and Eq.(8), respectively. However, because ODE considers a continuous-time regime, it is not obvious whether these results still hold in practical discrete-time implementations. The discrete-time performance guarantee has been shown in prior works (i.e., Theorem 4 of [1]). It would be beneficial if the performance guarantees of discrete-time GD and SGD algorithms can be provided, which is provided in prior work.

2. Theorem 3.2 considers the case where $\chi_1^* \chi_2^* \leq 0.5$. This assumption seems to be a bit restrictive because the constants $\chi_1^*$ and $\chi_2^*$ depend on the network itself. It looks that ADOM+ in [1] does not need such a restrictive assumption to ensure convergence. It would be helpful if an additional explanation can be provided to justify why such an assumption is required. It would also be beneficial to discuss whether similar assumptions are required to the mentioned benchmark algorithms.

3. Theorem 3.2 shows the existence of the parameters $\alpha,\gamma$ but does not specify how these parameters should be chosen in real networks. It would be helpful to specify the choice of these parameters. Also, does  $\gamma$ refer to the same quantity in Table 1?

4. Corollary 3.2.1 introduces an additional bias term $\frac{C_1}{\sqrt{\mu L}}$ to characterize the SGD system (8) in a continuous-time regime. It is claimed that "$L$ allows to adjust the trace-off bias-variance of the descent". This seems to be a bit confusing because $L$ represents the fixed Lipschitz constant. If one considers a larger $L$ to reduce this term, the decay rate of the term $C_0 e^{-ct\sqrt{\mu/L}}$ would deteriorate as well. It would be helpful if the authors could elaborate more on this.

5. When comparing DADAO with ADOM+, it is claimed that ADOM+ has potentially substantially higher expected communication than DADAO. It would be beneficial if the paper can specify the referred scenarios.

6. Various benchmark algorithms are tested in the numerical part. However, it seems that MSDA also has comparable performance with the proposed DADAO algorithm in most of the scenarios. Could the authors provide additional numerical experiments to demonstrate the efficiency of DADAO? It would be helpful if numerical experiments in some special cases such as star or complete networks can suggest DADAO's superior performance.

Minor Issues: There are some typos and some quantities are undefined.

1.  In the abstract, the $O(n \sqrt{\frac{L}{\mu}} \log \epsilon)$ gradient complexity is confusing. Should it be  $O(n \sqrt{\frac{L}{\mu}} \log \frac{ 1}{ \epsilon} )$ where $\epsilon$ is the precision?

2. In Table 1, $\gamma$ is undefined so that a fair comparison between $\sqrt{\chi_1\chi_2} n $ and $\chi_1 |\mathcal{E}|$ is not obvious.

3. When defining $\chi_2(t)$ in page 4, the quantity $\Lambda^+(t)$ is undefined.

References:
[1] Kovalev, D., Shulgin, E., Richtárik, P., Rogozin, A., and Gasnikov, A. (2021). ADOM: Accelerated decentralized optimization method for time-varying networks.

**Summary Of The Paper:**

This paper proposes a decentralized asynchronous stochastic first-order algorithm DADAO to minimize the sum of strongly convex functions over time-varying decentralized networks. This paper develops the theoretical performance guarantee of a related ODE system and tests the empirical performance by numerical simulations.

**Summary Of The Review:**

This paper proposes an algorithm to solve decentralized strongly convex optimization and establishes some new complexity results. The results are developed for continuing-time frameworks, but the guarantees for implementable practical frameworks are absent. The theoretical claims are also only applicable to a restrictive class of problems. The overall contribution seems to be limited.

---

> ### Author Response · Authors · 2022-11-09
> **Author response to reviewer gkNG (1/2)**
>
> We thank reviewer gkNG for the care they put into precisely stating the points of discussion they see. We now address each point of the reviewer successively:
>
> **Strength 1.** Not only are our performances “new,” but we are also happy to claim they are systematically better than any other prior work, as proved in Sec 4.1, Appendix F, and made explicit in Table 1 and Table 2. However, if the reviewer disagrees, we would be glad to understand which work outperforms our method.
>
> **Weakness 1.** We thank the reviewer for the distinction with a continuous setting. However, our work is rather *continuized*: system (3) is not an ODE but an SDE, obtained from a Pointwise Poisson Process. The corresponding dynamic is discrete, as observed in Alg. 1 of our paper. In other words, we have created a novel, randomized, discrete algorithm that relies on continuous proof. It is exactly the same framework as the last Neurips 2021 best paper [1]. Thus, our performance guarantees can be directly compared to previous work, particularly [1]!
>
> **Weakness 2.** Thanks: one can understand the condition $\chi_1 \chi_2 \leq 0.5$ as a condition on the minimum communication rate for the method to converge (see Appendix E). Thus, not only is this constraint actually in ADOM+ (in ADOM+, this constraint corresponds to the number of necessary gossip rounds (given page 9 of [2]) ), but it is also stronger: as $\chi_1 \vert \mathcal E \vert \geq n\sqrt{\chi_1 \chi_2 }$, it leads to a worse communication rate than ours. Please note that we already have this discussion in our paper: one of the strengths of our algorithm is to be flexible to the Laplacian matrix used, as stated in the first sentence of Sec 3: the condition $\sqrt{2\chi_1 \chi_2} \leq 1 $ is equivalent to multiplying the Laplacian matrix by those constants, which is stated in equation (5) and used everywhere in Sec. 4.1.
>
> **Weakness 3.** Thank you, we have added a reference in the statement of Theorem 3.2 to those constants, yet note those hyperparameters were already in Appendix H2, page 27 of our paper. In addition, we have made explicit all the constants of our theorem in order to reflect that we have just a standard bound for a sum of smooth and strongly convex functions. Thanks for the remark, we changed the notation for the spectral gap to $\rho$ to avoid confusions with the gradient step-size in our algorithm.
>
> **Weakness 4.** Thank you for pointing this: we have made explicit all the constants in this theorem in order to notice that we get the a term in $\frac{5\sigma^2}{\sqrt{\mu L}}$, which is standard in convex stochastic optimization! We would like to remind the reviewer that such constants are totally normal: for instance, they appear explicit in Theorem 7 of last year’s best paper at Neurips [1], and they also appear in [3,4,5,6,7,8]. Moreover, the extra variance term can be removed simply by considering a $L’>L$ (note that a $L$-smooth function is also $L’$-smooth): it affects the step size (and leads to a slightly longer convergence), while reducing the variance.
>
> **Weakness 5.** Thanks: not only the superiority of DADAO is stated in Table 1 and the scenarios of Table 2, but also note that the second paragraph of Sec 4.1 (whose proof is in Sec F.1) develops why our communications rate is better than ADOM+: the main reason is that ADOM+ requires $\chi_1|\mathcal{E}|$ communications due to the multi-consensus step (see page 9 of [2]). Our work needs only $\sqrt{\chi_1\chi_2}n$ communications, but: $\chi_2\leq \chi_1$ by definition and $n\leq|\mathcal{E}|$ by construction, which leads to better communication rates for DADAO. Figures 1-2 of the experiments and Appendix G.1 confirm our statement.
>
> **Weakness 6.** This is simply a matter of absolute constant: the scenario for the star graph in Table 2 shows theoretically that our performance is provably better. However, in order to verify our claim numerically, we ran an experiment (please note our code is linked to the appendix), which can be seen in Appendix G3: the figure clearly shows a constant factor between the gradient updates and, on the contrary, an increase in communication for MSDA in the order of $\sqrt{n}$. We hope the reviewer finds our experiments solidify our theoretical statement!

---

> > ### Author Response · Authors · 2022-11-09
> > **Author response to reviewer gkNG (2/2)**
> >
> > **Minor issue.**
> >
> > * Thanks! $f=\mathcal O(g)$ means that $|f| \leq C|g|$ for some positive constant $C$ (which is defined in part “Notation” of the paper of the Appendix), but we are happy to clarify this part: we replaced it everywhere with $\log \frac{1}{\epsilon}$
> > * Thanks, we have added a reference to Sec 4.1 (“Comparison with methods that depend on the spectral gap”) in Table 1 and an additional reference to Appendix F1 in Sec 4.13.
> > * Thanks! It was already in the notations, but we added for clarity the definition of the notation for the pseudo inverse in the main text before defining $\chi_2$, page 4.
> >
> > **Missing definitions:** We believe that everything has been well defined through all the paper (some of the notations are re-stated in “Notation” of the Appendix): however,we added the definition of some notations in the main text for clarity (even if we like to think of them as standard).
> >
> > **Summary of review:** We must stress that our method is implementable practically as is (and we have a source code!): Poisson Processes model the time each operation (computation or communication) takes, which, in practice, can be considered stochastic. And even if workers were unrealistically perfect, we could introduce stochasticity locally at each worker by sampling from an exponential law (which we do with our source code!).
> >
> > [1] Mathieu Even, Raphaël Berthier, Francis Bach, Nicolas Flammarion, Hadrien Hendrikx, Pierre Gaillard, Laurent Massoulié, and Adrien Taylor. A continuized view on Nesterov acceleration for stochastic gradient descent and randomized gossip. In Advances in Neural Information Processing Systems, 2021.
> >
> > [2] Dmitry Kovalev, Elnur Gasanov, Alexander Gasnikov and Peter Richtarik, Lower Bounds and Optimal Algorithms for Smooth and Strongly Convex Decentralized Optimization Over Time-Varying Networks, In Advances in Neural Information Processing Systems, 2021.
> >
> > [3]  Guanghui Lan. An optimal method for stochastic composite optimization. Math. Program., 2012.
> >
> > [4] Chonghai Hu, Weike Pan, and James Kwok. Accelerated gradient methods for stochastic optimization and online learning. In Advances in Neural Information Processing Systems, 2009.
> >
> > [5] Lin Xiao. Dual averaging methods for regularized stochastic learning and online optimization. J. Mach. Learn. Res., 2010.
> >
> > [6] Olivier Devolder. Stochastic first order methods in smooth convex optimization. Technical report, CORE, 2011.
> >
> > [7] Michael Cohen, Jelena Diakonikolas, and Lorenzo Orecchia. On acceleration with noisecorrupted gradients. In Proceedings of the 35th International Conference on Machine Learning, 2018.
> >
> > [8] Necdet Serhat Aybat, Alireza Fallah, Mert Gurbuzbalaban, and Asuman Ozdaglar. Robust accelerated gradient methods for smooth strongly convex functions. SIAM Journal on Optimization, 2020.

---

### Official Review · Reviewer_L3mL · 2022-10-31

**Confidence:** 3
**Correctness:** 3
**Technical Novelty And Significance:** 3
**Empirical Novelty And Significance:** Not applicable
**Recommendation:** 5

**Clarity, Quality, Novelty And Reproducibility:**

-Throughout the manuscript, underlying assumptions are clearly stated.
-I recommend moving Algorithm 2 from the appendix to the main manuscript.
-The clarity of convergence results requires some improvements. The main technical contribution of this paper lies in the gossip analysis. However, the optimization results are not presented and elaborated sufficiently. I recommend expanding the discussion in Subsection 3.3.


**Strength And Weaknesses:**

Strengths:
-The authors can achieve communication rounds and convergence complexity results by decoupling optimization (computation) from communication via two independent Poisson processes.
-The authors provide a clear statement of contributions and comparisons to the literature.
-A novel analysis of asynchronous communications via is provided in Section 3.

Weaknesses:
-The optimization results in Subsection 3.3 do not contain information on bounds for hyperparameters in the proposed dynamic in Subsection 3.2.
-The stochastic result (Corollary 3.2.1) shows a non-vanishing error in the rate. It is conventional to tweak the hyperparameters (e.g., a decreasing learning rate) to make this term vanish over time. It is unclear why this term is a constant and if it can be improved. The current format undermines stochastic optimization results.
-Numerical comparisons are limited to the ADOM+ algorithm. I recommend further comparisons with other algorithms, such as Gradient Tracking.


**Summary Of The Paper:**

The authors study the decentralized asynchronous optimization problem on a time-varying network with n nodes where the main purpose is to minimize the sum of  L-smooth and \mu-strongly convex functions distributed on the nodes. The authors decouple the computation and communication steps in the asynchronous optimization problem by modeling this problem as a union of two independent Poisson Point Processes and consider primal gradients in their proposed method. They show convergence results for this problem under the proper assumptions. They also present numerical results corroborating their findings in the convergence results.


**Summary Of The Review:**

A new study on decentralized asynchronous optimization on time-varying graphs is made in this paper. By modeling the problem via Poisson processes, the authors present communication and computation complexity for their proposed algorithm. Some improvements in Subsection 3.3 are required to clarify the findings for stochastic optimization.

---

> ### Author Response · Authors · 2022-11-08
> **Author response to reviewer L3mL**
>
> We thank reviewer L3mL for the valuable feedback and are glad they found our statements clear and novel. We would also like to re-emphasize that our work is the only work that enjoys state-of-the-art computation & communication convergence rates in the context of decentralized-asynchronous optimization (see Table 1, Table 2) but also uses the effective resistance instead of the spectral gap! We now answer each comment in the order of the text.
>
> * Thanks for the suggestion! We have added a reference to the already existing Appendix H2 of our paper in  Theorem 3.2 to clarify the dependencies of our parameters. We also made explicit all the constants in Theorem 3.2 and its corollary.
> * Thanks for raising this point! We have rephrased our Theorem 3.2 and its SGD Corollary to clarify all the constants so that the reviewer recovers the standard and classic SGD rate under additive noise for a strongly convex and smooth function. We would like to remind the reviewer that such constants are totally normal: for instance, they appear explicit in Theorem 7 of last year’s best paper at Neurips [1], and they also appear in [2,3,4,5,6,7]. Moreover, for “tweaking” the learning rate, it is conventional to consider $L’>L$ as a parameter of the algorithm to reduce this variance term and guarantee $\epsilon$ precision. We hope the new format of our theorem statement pleases the reviewer.
> * Thanks for pointing out those numerical issues! We would like to remind you that we also compare to MSDA, which relies on the spectral gap: we added an experiment in Appendix G3 which shows that our rates are better on a star graph than either work based on $|\mathcal{E}|\chi_1^*$ and either work based on $|\mathcal{E}|\sqrt{\rho^*}$ (where $\rho^*$ is the spectral gap): the gradient tracking falls in those categories (see the “Accelerated Gradient Tracking” entry in Table 1). Consequently, as we proved that our method is uniformly better than concurrent work, that we tested each possible type of complexity numerically, and that Table 2 provides some scenarios where our method is provably better, we find the comparisons accurate. If the reviewer suggests a specific class of algorithms with a different complexity that we have forgotten, we would happily compare our work to it! Note also that our method enjoys time-varying settings, asynchrony, and decoupling, which is slightly non-trivial to obtain.
> * Thanks for the suggestion! Due to the lack of space, we only added a reference to Algorithm 2, close to Algorithm 1.
> * Concerning the presentation: we stated all constants and results explicitly in theorem 3.2 in a format identical to [1], which should clarify all the reviewer's concerns. However, our contribution is not limited to a “gossip analysis.” On the contrary, the finding of a novel Lyapunov function in conjunction with a novel SDE (see Sec 3.2 and 3.3) allows our method to enjoy all its advantages.
>
> Thanks again for your reviewing time.
>
> [1] Mathieu Even, Raphaël Berthier, Francis Bach, Nicolas Flammarion, Hadrien Hendrikx, Pierre Gaillard, Laurent Massoulié, and Adrien Taylor. A continuized view on Nesterov acceleration for stochastic gradient descent and randomized gossip. In Advances in Neural Information Processing Systems, 2021.
>
> [2]  Guanghui Lan. An optimal method for stochastic composite optimization. Math. Program., 2012.
>
> [3] Chonghai Hu, Weike Pan, and James Kwok. Accelerated gradient methods for stochastic optimization and online learning. In Advances in Neural Information Processing Systems, 2009.
>
> [4] Lin Xiao. Dual averaging methods for regularized stochastic learning and online optimization. J. Mach. Learn. Res., 2010.
>
> [5] Olivier Devolder. Stochastic first order methods in smooth convex optimization. Technical report, CORE, 2011.
>
> [6] Michael Cohen, Jelena Diakonikolas, and Lorenzo Orecchia. On acceleration with noisecorrupted gradients. In Proceedings of the 35th International Conference on Machine Learning, 2018.
>
> [7] Necdet Serhat Aybat, Alireza Fallah, Mert Gurbuzbalaban, and Asuman Ozdaglar. Robust accelerated gradient methods for smooth strongly convex functions. SIAM Journal on Optimization, 2020.

---

### Author Response · Authors · 2022-11-08
**General comment for the reviewers**

We thank the reviewers for their helpful comments in clarifying some points of this paper. Before answering the reviewer, we would like to re-emphasize that:

* DADAO obtains state-of-the-art rates in computations and communications while allowing more flexibility than its concurrent works (in asynchronous and time-varying graph settings).
* Several reviewers pointed out our algorithm to be a “continuous ODE.” However, we are concerned because DADAO uses a pointwise Poisson process, which has to be understood as an SDE: this is more detailed in the proof technique introduced last year, see [1].
* Some reviewers ask for a “numerical proof”: we have added an extra experiment for the star graph (in Appendix G3), but we would like to emphasize that Table 2 proposes several scenarios in which DADAO is provably better than its concurrent works. Furthermore, we, unfortunately, do not understand what is a “more complex model” in a strongly convex and smooth setting: we follow the standard approach (see, e.g. [2,3,4]) of studying the distributed linear and regularized logistic regression.

To simplify the reviewer's work, we added our modification in red. Finally, note that we restated our main theorem with all explicit constants for clarity and thank the reviewers for this suggestion.


[1] Mathieu Even, Raphaël Berthier, Francis Bach, Nicolas Flammarion, Hadrien Hendrikx, Pierre Gaillard, Laurent Massoulié, and Adrien Taylor. A continuized view on nesterov acceleration for stochastic gradient descent and randomized gossip. In Advances in Neural Information Processing Systems, 2021.

[2] Anastasia Koloskova, Nicolas Loizou, Sadra Boreiri, Martin Jaggi, Sebastian Stich. A Unified Theory of Decentralized SGD with Changing Topology and Local Updates. In Proceedings of the 37th International Conference on Machine Learning, 2020.

[3] Huan Li and Zhouchen Lin. Accelerated gradient tracking over time-varying graphs for decentralized optimization. arXiv preprint arXiv:2104.02596, 2021.

[4] Dmitry Kovalev, Elnur Gasanov, Alexander Gasnikov and Peter Richtarik, Lower Bounds and Optimal Algorithms for Smooth and Strongly Convex Decentralized Optimization Over Time-Varying Networks, In Advances in Neural Information Processing Systems, 2021.

---

### Decision · Program_Chairs · 2023-01-20

**Decision:**

Reject

**Justification For Why Not Higher Score:**

NA

**Justification For Why Not Lower Score:**

NA

**Metareview: Summary, Strengths And Weaknesses:**

This work considers a decentralized optimization problem, and develops an asynchronous accelerated gossip algorithm to solve it. The strengths are in providing theoretical guarantees and the innovation of using Poisson processes to model the asynchronous time-stamps at which algorithmic updates are executed. The algorithmic approach is fairly similar to prior works, meaning the main contrast is in employing  the Poisson counters for modeling the asynchronous updates. The consensus opinion of the reviewers is that this is insufficiently innovative to meet the bar of novelty required for acceptance. I tend to agree that this problem is so well-studied, additional efforts must provide a substantial departure so as to not retread past works. That this is the case is evidenced by the diversity of references called out by the reviewers, which have minimal overlap with one another. Unfortunately, this paper does not qualify as a substantial depature.